# Cell-wall remodeling drives engulfment during *Bacillus subtilis* sporulation

**Nikola Ojkic[1,2†], Javier López-Garrido[3†], Kit Pogliano[3*], Robert G Endres[1,2*]**

[1]Department of Life Sciences, Imperial College London, London, United Kingdom; [2]Centre for Integrative Systems Biology and Bioinformatics, Imperial College London, London, United Kingdom; [3]Division of Biological Sciences, University of California, San Diego, La Jolla, United States

**Abstract** When starved, the Gram-positive bacterium *Bacillus subtilis* forms durable spores for survival. Sporulation initiates with an asymmetric cell division, creating a large mother cell and a small forespore. Subsequently, the mother cell membrane engulfs the forespore in a phagocytosis-like process. However, the force generation mechanism for forward membrane movement remains unknown. Here, we show that membrane migration is driven by cell wall remodeling at the leading edge of the engulfing membrane, with peptidoglycan synthesis and degradation mediated by penicillin binding proteins in the forespore and a cell wall degradation protein complex in the mother cell. We propose a simple model for engulfment in which the junction between the septum and the lateral cell wall moves around the forespore by a mechanism resembling the 'template model'. Hence, we establish a biophysical mechanism for the creation of a force for engulfment based on the coordination between cell wall synthesis and degradation.

**\*For correspondence:**
kpogliano@ucsd.edu (KP);
r.endres@imperial.ac.uk (RGE)

[†]These authors contributed equally to this work

**Competing interests:** The authors declare that no competing interests exist.

## Introduction

To survive starvation, the Gram-positive bacterium *Bacillus subtilis* forms durable endospores (*Tan and Ramamurthi, 2014*). The initial step of sporulation is the formation of an asymmetrically positioned septum (polar septation), which produces a larger mother cell and a smaller forespore (*Figure 1A*). After division, the mother cell engulfs the forespore in a phagocytosis-like manner. Engulfment entails a dramatic reorganization of the sporangium, from two cells that lie side by side to a forespore contained within the cytoplasm of the mother cell. The internalized forespore matures and is ultimately released to the environment upon mother cell lysis. After engulfment, the forespore is surrounded by two membranes within the mother cell cytoplasm, sandwiching a thin layer of peptidoglycan (PG) (*Tocheva et al., 2013*). While a number of molecular players for engulfment have been identified, the mechanism of force generation to push or pull the mother cell membrane around the forespore remains unknown (*Higgins and Dworkin, 2012*).

The cellular machinery for engulfment is complex, presumably to add robustness for survival (*Figure 1A*, inset). First, the forespore protein SpoIIQ and the mother cell protein SpoIIIAH interact in a zipper-like manner across the septum (*Blaylock et al., 2004*), and mediate the fast engulfment observed in the absence of cell wall (*Broder and Pogliano, 2006*; *Ojkic et al., 2014*). This complex is static and is proposed to act as a Brownian ratchet to prevent backwards movement of the engulfing membrane, contributing to the robustness of engulfment in intact cells (*Sun et al., 2000*; *Broder and Pogliano, 2006*). Second, the SpoIID, SpoIIM and SpoIIP complex (DMP) localizes at the leading edge (LE) of the mother cell engulfing membrane and is essential and rate limiting for membrane migration (*Abanes-De Mello et al., 2002*; *Gutierrez et al., 2010*). The complex contains two enzymes that degrade PG in a processive manner: SpoIIP removes stem peptides, and SpoIID degrades the resulting denuded glycan strands (*Abanes-De Mello et al., 2002*; *Chastanet and*

**eLife digest** Some bacteria, such as *Bacillus subtilis,* form spores when starved of food, which enables them to lie dormant for years and wait for conditions to improve. To make a spore, the bacterial cell divides to make a larger mother cell and a smaller forespore cell. Then the membrane that surrounds the mother cell moves to surround the forespore and engulf it. For this process to take place, a rigid mesh-like layer called the cell wall, which lies outside the cell membrane, needs to be remodelled. This happens once a partition in the cell wall, called a septum, has formed, separating mother and daughter cells. However, it is not clear how the mother cell can generate the physical force required to engulf the forespore under the cramped conditions imposed by the cell wall.

To address this question, Ojkic, López-Garrido et al. used microscopy to investigate how *B. subtilis* makes spores. The experiments show that, in order to engulf the forespore, the mother cell must produce new cell wall and destroy cell wall that is no longer needed. Running a simple biophysical model on a computer showed that coordinating these two processes could generate enough force for a mother cell to engulf a forespore.

Ojkic, López-Garrido et al. propose that the junction between the septum and the cell wall moves around the forespore to make room for the mother cell's membrane for expansion. Other spore-forming bacteria that threaten human health – such as *Clostridium difficile,* which causes bowel infections, and *Bacillus anthracis,* which causes anthrax – might form their spores in the same way, but this remains to be tested. More work will also be needed to understand exactly how bacterial cells coordinate the cell wall synthesis and cell wall degradation.

*Losick, 2007*; *Morlot et al., 2010*; *Gutierrez et al., 2010*). Mutants with reduced SpoIID or SpoIIP activity or protein levels engulf asymmetrically, with the engulfing membrane migrating faster on one side of the forespore (*Abanes-De Mello et al., 2002*; *Gutierrez et al., 2010*). Third, blocking PG precursor synthesis with antibiotics impairs membrane migration in mutants lacking the Q-AH zipper, suggesting that PG synthesis at the LE of the engulfing membrane contributes to engulfment (*Meyer et al., 2010*; *Tocheva et al., 2013*). However, the mechanistic details of membrane migration and for the coordination between PG synthesis and degradation remain unclear.

The biophysical principles of cell wall remodeling in Gram-positive bacteria are not well understood. In *Bacillus subtilis*, the cell wall is about 20–40 nm thick, and is likely organized into multiple (20–30) PG layers (*Morlot et al., 2010*; *Reith and Mayer, 2011*; *Lee and Huang, 2013*; *Misra et al., 2013*; *Dover et al., 2015*). In contrast, cryo-electron tomography has demonstrated that a thin PG layer is present between the septal membranes throughout engulfment, appearing to form continuous attachments with the old cell wall (*Tocheva et al., 2011*, *2013*). The outer cell wall of Gram-positive bacteria also contains a significant amount of teichoic acids, important for cell morphology, phosphates, and antibiotic resistance (*Grant, 1979*; *Brown et al., 2013*) but largely absent in spores (*Chin et al., 1968*; *Johnstone et al., 1982*). Engulfment entails extensive cell wall remodeling, with peptidoglycan precursors, newly synthesized PG and the sporulation specific PG degradation machinery localizing at the LE of the engulfing membrane (*Meyer et al., 2010*; *Tocheva et al., 2013*; *Abanes-De Mello et al., 2002*). However, since engulfment occurs at high turgor pressure within the cramped confines of the thick outer cell wall, we expect that membrane movement is severely reduced by steric hindrance (*Lizunov and Zimmerberg, 2006*). Hence, we anticipate that peptidoglycan remodeling is a critical step in engulfment, which may either act as a force generator or simply create room for engulfment by the mother cell membrane.

Here, we provide a biophysical mechanism for engulfment in which PG synthesis and degradation move the junction between the septal PG and the lateral cell wall around the forespore, making room for the engulfing membrane to move by entropic forces. Using antibiotics that block different steps in PG synthesis, we demonstrate that PG synthesis is essential for membrane migration in all conditions and contributes to the localization of SpoIIDMP at the LE. We also show that components of the PG biosynthetic machinery, including several penicillin binding proteins (PBPs) and the actin-like proteins MreB, Mbl and MreBH track the LE of the engulfing membrane when produced in the

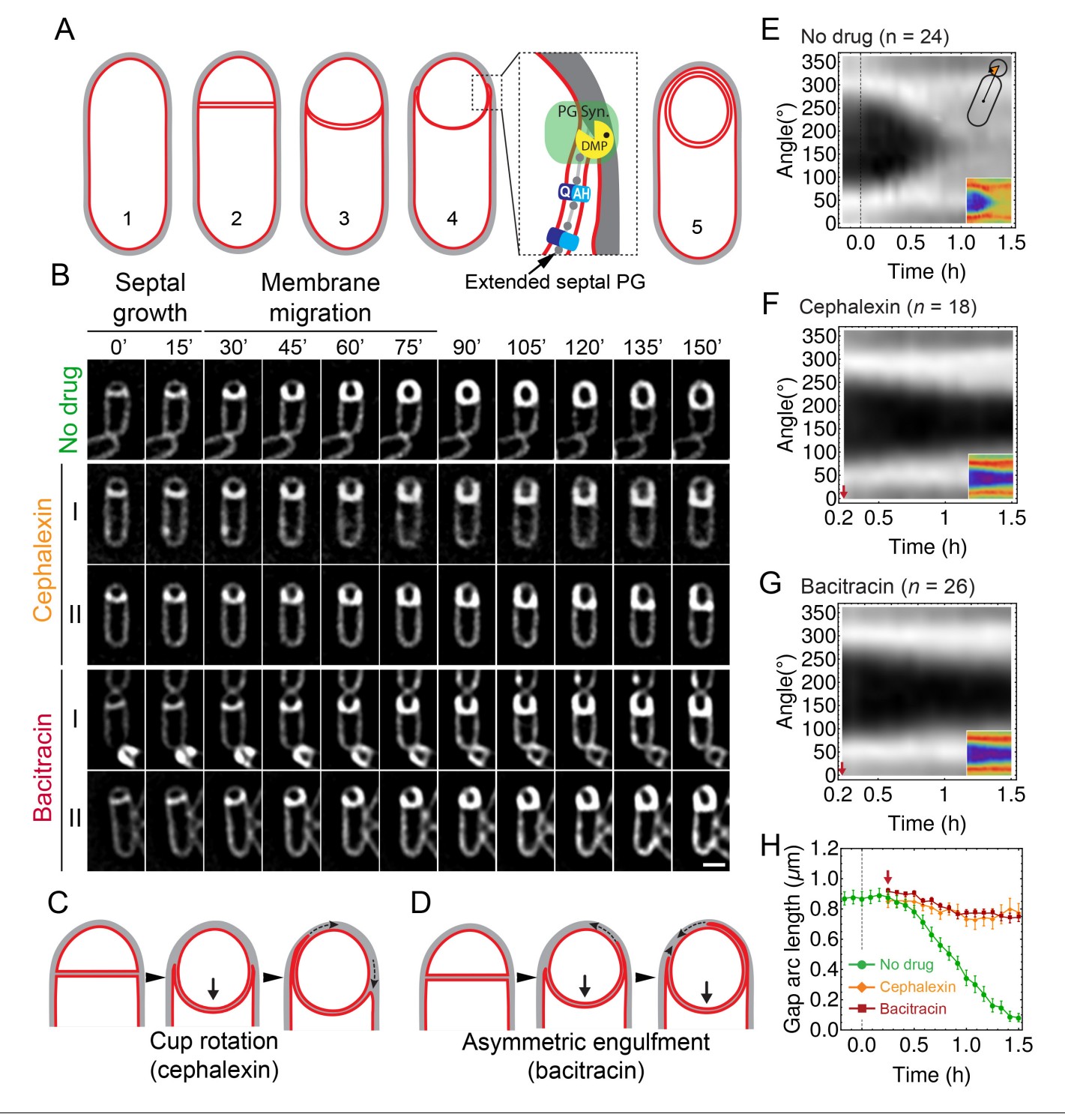

**Figure 1.** Peptidoglycan (PG) synthesis is essential for leading-edge (LE) migration. (**A**) Morphological changes during spore formation. Peptidoglycan shown in grey, membrane in red. (1) Vegetative cell. (2) The first morphological step in sporulation is asymmetric cell division, producing a smaller forespore and a larger mother cell. (3) The septum curves and protrudes towards the mother cell. (4) The mother cell membrane migrates towards the forespore pole. The different modules contributing to membrane migration are shown in the inset (see Introduction for details). During engulfment, the septal PG is extended around the forespore (**Tocheva et al., 2013**). (5) Fully engulfed forespore surrounded by two membranes sandwiching a thin layer of PG. (**B**) Snapshots of engulfing sporangia from time-lapse movies in the absence of antibiotics, or in the presence of cephalexin or bacitracin. Cells were stained with fluorescent membrane dye FM 4–64 and imaged in medial focal plane. In the absence of antibiotics (top) the septum curves and grows towards the mother cell without significant forward movement of the engulfing membrane for ~20 min. After that, the LE of the engulfing

*Figure 1 continued on next page*

*Figure 1 continued*

membrane starts migrating and reaches the forespore pole in ~1 hr. When PG precursor delivery system is blocked with bacitracin (50 μg/ml): (I) LE migration is stopped or (II) engulfment proceeds asymmetrically. Similar results are obtained when cells are treated with cephalexin (50 μg/ml). However, in this case the asymmetric engulfment phenotype observed at later time points is due to rotation of the engulfment cup (**C**) rather than to asymmetric movement forward of the engulfing membrane (**D**). (**E**) FM 4–64 average kymograph of *n* = 24 engulfing cells (see Materials and methods, Appendix 1). Average fluorescent intensity along forespore contour vs time in the mother-forespore reference frame as shown in top inset. All cells are aligned in time based on time 0' (0 min). Time 0' is assigned to the onset of curving septum (*Figure 1—figure supplement 3*). Bottom inset is average kymograph represented as heat map. (**F–G**) Average kymograph for cells treated with cephalexin (*n* = 18) (**F**) or bacitracin (*n* = 26). (**G**) When drug was added analyzed cells had (55 ± 5)% engulfment (red arrow). The percentage of engulfment is calculated as total angle of forespore covered with mother membrane divided by full angle. All cells had fully curved septum. Non-engulfed part of the forespore is represented as the black regions in kymographs. (**H**) In untreated sporangia, gap starts to close ~20 min after onset of membrane curving. In antibiotic-treated cells gap does not close. Sample size as in (**F–G**). Red arrow points when drug is added. Average ± SEM. Scale bar 1 μm.

The following figure supplements are available for figure 1:

**Figure supplement 1.** Sporulation minimal inhibitory concentration.

**Figure supplement 2.** Quantification of cell division events in timelapse movies.

**Figure supplement 3.** Image analysis of non-treated cells.

forespore, but not when produced in the mother cell. We implement a biophysical model for PG remodeling at the LE of the engulfing membrane, based on the 'template mechanism' of vegetative cell growth and implemented by stochastic Langevin simulations. These simulations reproduce experimentally observed engulfment dynamics, forespore morphological changes, and asymmetric engulfment when PG synthesis or degradation is perturbed. Taken together, our results suggest that engulfment entails coordination of PG synthesis and degradation between the two compartments of the sporangium, with forespore-associated PBPs synthesizing PG ahead of the LE and the mother-cell DMP complex degrading this PG to drive membrane migration.

## Results

### PG synthesis is essential for membrane migration

In contrast to previous studies (*Meyer et al., 2010*), we attempted to find conditions that completely blocked PG synthesis in sporulating cultures (*Figure 1—figure supplement 1*). To estimate the sporulation minimal inhibitory concentration (sMIC) of antibiotics, we monitored the percentage of cells that had undergone polar septation over time in batch cultures. Polar septation depends on PG synthesis and is easy to track visually (*Pogliano et al., 1999*), which makes it a good indicator for efficient inhibition. We assayed nine antibiotics inhibiting different steps in the PG biosynthesis pathway, and found concentrations that blocked the formation of new polar septa for seven of them (*Figure 1—figure supplement 1B,C*). In most cases, the antibiotic concentration that blocked polar septation also inhibited completion of engulfment (*Figure 1—figure supplement 1B*). Only two drugs, fosfomycin and D-cycloserine, failed to completely block polar cell division. These drugs inhibit production of PG precursors that, during starvation conditions, might be obtained by recycling rather than *de novo* synthesis (*Reith and Mayer, 2011*), potentially from cells that lyse during sporulation, as has been observed in studies of

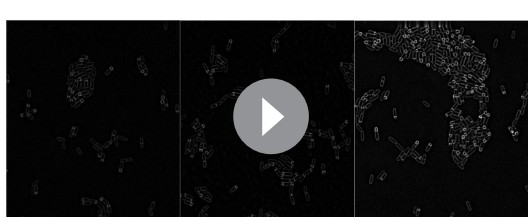

**Video 1.** Timelapse microscopy of sporulating *B. subtilis* stained with the membrane dye FM 4–64. The left panel shows untreated cells, the middle panel cephalexin-treated cells (50 μg/ml), and the right panel bacitracin-treated cells (50 μg/ml). Cells were imaged in agarose pads supplemented with the appropriate antibiotics (see Materials and methods for details). Pictures were taken every 5 min. Total time 2.5 hr.

*B. subtilis* cannibalism (*González-Pastor et al., 2003*; *Straight and Kolter, 2009*; *Lamsa et al., 2012*), or from cells that lyse due to antibiotic treatment (*Lamsa et al., 2016*). These results demonstrate that the later stages in PG synthesis are essential for engulfment in wild type sporangia.

To investigate the role played by PG synthesis, we selected two antibiotics for further characterization: cephalexin, which inhibits PBP activity, and bacitracin, which blocks cell-wall precursor delivery (recycling of undecaprenyl phosphate). Using time-lapse microscopy (see Materials and methods for details), we monitored membrane dynamics during engulfment in the medial focal plane using the fluorescent membrane dye FM 4–64 (*Figure 1B*, *Video 1*). In these 2–5 hour-long movies we observed occasional cell division events occurred with bacitracin (0.08 division events/cell after 90 min, compared to 0.28 division events/cell in untreated cultures, *Figure 1—figure supplement 2*), indicating that PG synthesis was not completely blocked under these conditions. However, negligible cell divisions occurred with cephalexin, indicating that PG synthesis was indeed completely blocked (*Figure 1—figure supplement 2*).

To better monitor LE dynamics we used two image analysis approaches (see Materials and methods for details). First, we created kymographs along forespore membranes (*Figure 1E–G*). The angular position of forespore pixels was calculated relative to the mother-forespore frame of reference (*Figure 1E*, inset). All cells were aligned in time based on the onset of septum curving (*Figure 1—figure supplement 3*), and for a given angle, the average fluorescence of different cells was calculated and plotted over time. Second, we measured the decrease in the distance between the two LEs of the engulfing membrane in the focal plane (the gap arc length), in order to directly assess movement of the LE around the forespore (*Figure 1H*).

These analyses demonstrated that in untreated sporangia (*Figure 1B*, top row), the septum curves and the forespore grows into the mother cell without significant forward movement of the LE for ~20 min after polar septation (at 30°C, *Figure 1H*). Subsequently, the LE of the engulfing membrane moves towards the forespore pole and engulfment completes within ~60 min (*Figure 1E,H*). In sporangia treated with cephalexin (*Figure 1B*, middle row I), the septum curves and extends towards the mother cell, but there is no forward membrane migration (*Figure 1F,H*). Sometimes the LE retracted on one side while advancing slightly on the other (typically occurred after 90 min of imaging; *Figure 1B*, middle row II), which appears to be the rotation of the 'cup' formed by the engulfing membranes relative to the lateral cell wall (*Figure 1C*).

Similar to cephalexin, in most sporangia treated with bacitracin (*Figure 1B*, bottom row I), the forespore extended into the mother cell without significant membrane migration (*Figure 1G,H*). However, in ~20% of the sporangia, the engulfing membrane migrated asymmetrically, with one side moving faster than the other, although usually it failed to completely surround the forespore (*Figure 1B*, bottom row II; *Figure 1D*). The continued engulfment under bacitracin treatment might be related to the fact that PG synthesis is not completely blocked in bacitracin-treated cells under time-lapse conditions (*Figure 1—figure supplement 2*). Taken together, these results suggest that PG synthesis is not only essential for the final stage of engulfment (membrane fission) in wild type cells (*Meyer et al., 2010*), but also for migration of the LE of the engulfing membrane around the forespore.

## PBPs accumulate at the leading edge of the engulfing membrane

It has been previously shown that there is an accumulation of membrane-bound PG precursors at the LE of the engulfing membrane (*Meyer et al., 2010*). Furthermore, staining with fluorescent D-amino acids has demonstrated that new PG is synthesized at or close to the LE (*Tocheva et al., 2013*). To investigate if there is a concomitant accumulation of PBPs at the LE, we stained sporangia with BOCILLIN-FL, a commercially available derivative of penicillin V that has a broad affinity for multiple PBPs in *B. subtilis* (*Lakaye et al., 1994*; *Zhao et al., 1999*; *Kocaoglu et al., 2012*). We observed continuous fluorescent signal around the mother cell membrane that was enriched at the LE (*Figure 2A*). To better monitor localization of PBPs during engulfment, we plotted fluorescence intensities along the forespores for the membrane and BOCILLIN-FL fluorescent signals as a function of the engulfment stage (*Figure 2B*). Clearly, the LE is always enriched with PBPs throughout membrane migration.

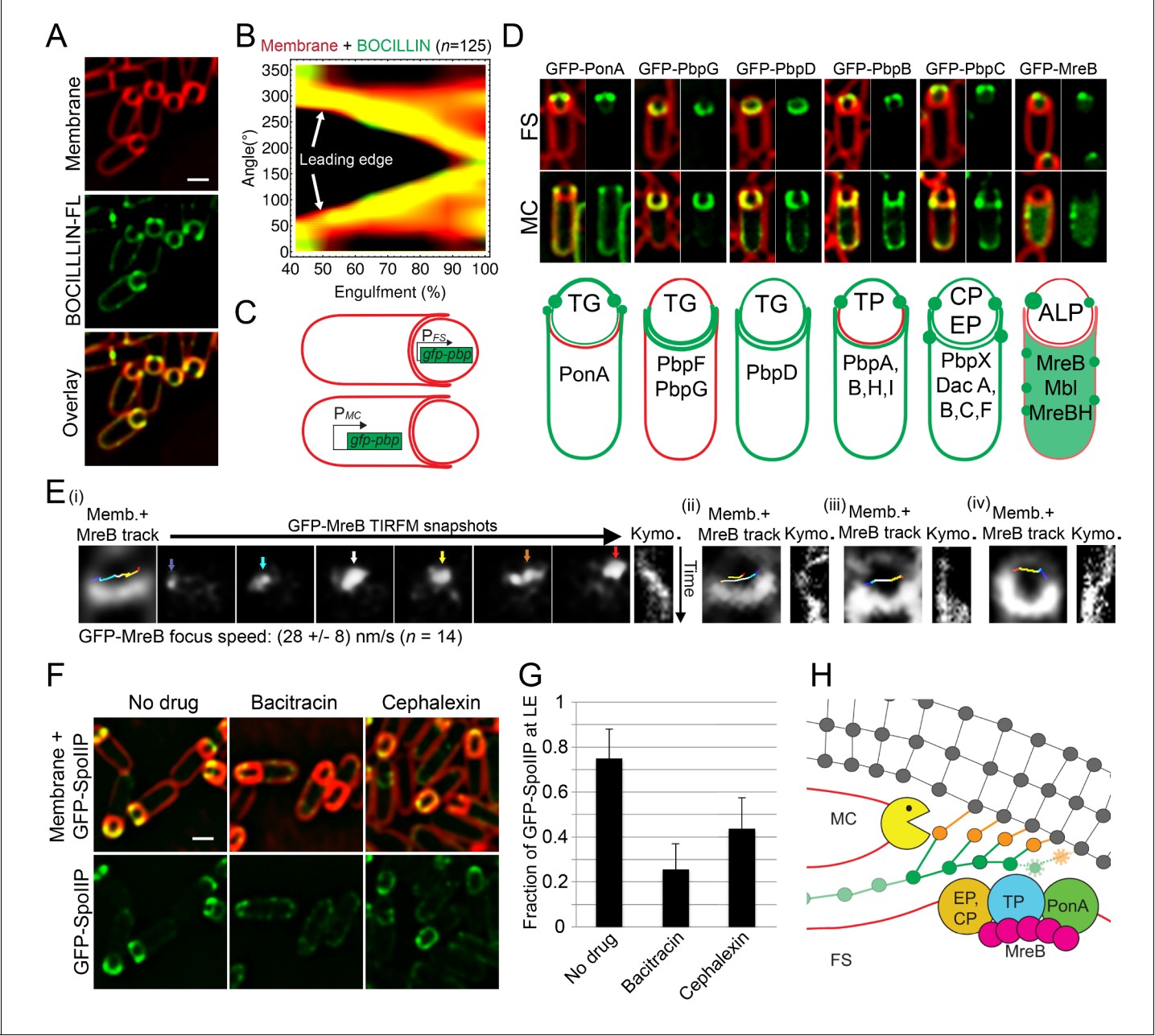

**Figure 2.** PG synthesis at the LE of the engulfing membrane by forespore PBPs contribute to proper localization of the DMP complex. (**A**) Sporulating cells stained with a green fluorescent derivative of penicillin V (BOCILLIN-FL). Bright foci are observed at the LE of the engulfing membrane. Membranes were stained with FM 4–64 (red). (**B**) Average BOCILLIN-FL (green) and FM 4–64 (red) fluorescence intensities along forespore contours plotted as a function of the degree of engulfment. Cells are binned according to percentage of engulfment. BOCILLIN-FL signal is enriched at the LE throughout engulfment (*n* = 125). (**C**) Cell-specific localization of the peptidoglycan biosynthetic machinery. GFP tagged versions of different *B. subtilis* PBPs and actin-like proteins (ALPs) were produced from mother cell- (MC) or forespore- (FS) specific promoters. (**D**) Six different localization patterns were observed upon cell-specific localization of PBPs and ALPs. For each pair of images, left panel shows overlay of membrane and GFP fluorescence, while the right panel only shows GFP fluorescence. Pictures of representative cells displaying the different patterns are shown (top, GFP fusion proteins transcribed from spoIIR promoter for forespore-specific expression, and from spoIID promoter for mother cell-specific expression). The six different patterns are depicted in the bottom cartoon and proteins assigned to each one are indicated. Membranes were stained with FM 4–64. See *Figure 2—figure supplement 1* for cropped fields of all PBPs we assayed. Transglycosylase (TG), transpetidase (TP), carboxipetidase (CP), endopeptidase (EP), actin-like protein (ALP). (**E**) TIRF microscopy of forespore-produced GFP-MreB in four different forespores (i to iv). In every case, the leftmost picture is an overlay of the forespore membranes (shown in white) and the tracks followed by individual TIRF images of GFP-MreB (color encodes time, from blue to red). Sporangia are oriented with the forespores up. For the first sporangia (i), snapshots from TIRF timelapse experiments taken 8 s apart are shown.
*Figure 2 continued on next page*

*Figure 2 continued*

Arrows indicate GFP-MreB foci and are color coded to match the trace shown in the left panel. Rightmost panel for each forespore shows a kymograph representing the fluorescence intensity along the line joining the leading edges of the engulfing membrane over time (from top to bottom; total time 100 s). Average focus speed (n = 14) is indicated at the bottom. Timelapse movies of the examples presented here and additional sporangia are shown in *Video 2*. (F) Localizaiton of GFP-SpoIIP in untreated sporangia, or in sporangia treated with bacitracin (50 µg/ml) or cephalexin (50 µg/ml). (G) Fraction of GFP-SpoIIP fluorescence at LE of the engulfing membrane. Bars represent the average and standard error of 85 untreated sporangia, 38 sporangia treated with bacitracin (50 µg/ml), and 67 sporangia treated with cephalexin (50 µg/ml). (H) Model for PG synthesis and degradation at the LE of the engulfing membrane. New PG is synthesized ahead of the LE of the engulfing membrane by forespore-associated PG biosynthetic machinery, and is subsequently degraded but the mother-cell DMP complex. We propose that DMP has specificity for the peptide cross-links that join the newly synthesized PG with the lateral cell wall (orange), which leads to the extension of the septal PG around the forespore. Scale bars 1 µm.

The following figure supplements are available for figure 2:

**Figure supplement 1.** Cell-specific localization of PBPs and actin-like proteins.

**Figure supplement 2.** Localization of forespore GFP-PonA and GFP-PbpA in different mutant backgrounds.

**Figure supplement 3.** SpoIIDMP localization upon treatment with different antibiotics blocking PG synthesis.

## PG biosynthetic machinery tracks the leading edge of the engulfing membrane from the forespore

One possible explanation for the requirement of PG synthesis for engulfment is that PG polymerization by PBPs associated with the LE of the engulfing membrane creates force to pull the engulfing membrane around the forespore. If so, we would expect the PBPs to be located in the mother cell membrane as they polymerize PG. To test this possibility, we assessed the localization of components of the PG biosynthetic machinery in the mother cell or forespore by producing GFP-tagged fusion proteins from promoters that are only active in the mother cell (P$_{spoIID}$) or in the forespore (the stronger P$_{spoIIQ}$ and the weaker P$_{spoIIR}$) after polar septation (*Figure 2C,D*, *Figure 2—figure supplement 1*). One prior study tested the localisation of several PBPs during sporulation (*Scheffers, 2005*), but most of them were produced before polar separation and it was not possible to determine which cell compartment they were in. We successfully determined the cell-specific localization of 16 proteins involved in PG synthesis (*Figure 2—figure supplement 1*), including all class A and four class B high-molecular-weight (HMW) PBPs, five low-molecular-weight (LMW) PBPs (four endopeptidases and one carboxipeptidase), and all three MreB paralogues (actin-like proteins, ALPs). Surprisingly, only PonA (PBP1a/b) showed a weak enrichment at the LE of the engulfing membrane when produced in the mother cell (*Figure 2D*). However, ten PBPs, including PonA and all the class B HMW PBPs and LMW PBPs tested, and all the MreB paralogues were able to track the LE only when produced in the forespore (*Figure 2D*, *Figure 2—figure supplement 1*). To follow the dynamics of the forespore PG biosynthetic machinery at the LE, we monitored the movement of GFP-MreB using TIRF microscopy (*Garner et al., 2011*; *Domínguez-Escobar et al., 2011*). Forespore GFP-MreB foci rotate around the forespore, coincident with the leading edge of the engulfing membrane, with speeds consistent with those previously reported (*Figure 2E*, *Video 2*).

It is unclear how the PBPs recognize the LE, as localization of forespore produced GFP-PonA and GFP-PbpA did not depend on candidate proteins SpoIIB, SpoIID, SpoIIM, SpoIIP, SpoIIQ, SpoIIIAH, SpoIVFAB, or GerM (*Aung et al., 2007*; *Abanes-De Mello et al., 2002*; *Chastanet and Losick, 2007*; *Blaylock et al., 2004*; *Rodrigues et al., 2016*) (*Figure 2—figure supplement 2*). However, these results indicate that the forespore plays a critical role in PG synthesis, and point to an engulfment mechanism

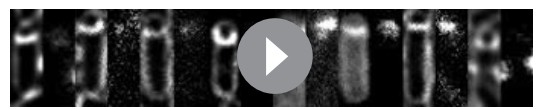

**Video 2.** Circumferential movement of forespore GFP-MreB. The movie shows the movement forespore GFP-MreB in eight different sporangia, determined by TIRF microscopy. A static membrane picture is shown to the left, and the TIRF microscopy of the corresponding GFP-MreB is shown immediately to the right. TIRF pictures were taken every 4 s, and the total duration of the movie is 100 s. The first four sporangia correspond to the examples (i) to (iv) shown in *Figure 2*.

that does not depend on pulling the engulfing membrane by mother cell-directed peptidoglycan synthesis.

## PG synthesis is required for SpoIIDMP localization at the leading edge of the engulfing membrane

The observation that multiple PBPs can track the LE of the engulfing membrane from the forespore opens the possibility that PG synthesis happens ahead of the LE, preceding PG degradation by the mother cell DMP complex. In this context, PG synthesis might be required for proper activity and/or localization of the DMP complex, which is the only other essential engulfment module described so far. The DMP complex localizes at the LE throughout engulfment (*Gutierrez et al., 2010*). To determine if PG synthesis is required for proper localization of DMP, we studied the localization of a GFP-SpoIIP fusion protein when PG synthesis was inhibited by different antibiotics (*Figure 2F,G*). GFP-SpoIIP shows a well-defined localization at the LE, with ~70% of the total GFP fluorescence at LE in native conditions (*Figure 2F,G*). However, when PG biosynthesis is inhibited, there is a delocalization of GFP-SpoIIP, which is almost total in cells treated with bacitracin and partial when antibiotics targeting later stages of PG synthesis are used (*Figure 2F,G*; *Figure 2—figure supplement 3*). Equivalent results were obtained with GFP-SpoIID and GFP-SpoIIM fusions (*Figure 2—figure supplement 3*). These results are consistent with a model in which PG is synthesized ahead of the LE by forespore-associated PBPs specify the site of PG degradation by the DMP complex (*Figure 2H*).

## A biophysical model to describe leading edge migration

Our data indicate that engulfment proceeds through coordinated PG synthesis and degradation at the LE. To explain how this coordination leads to engulfment, we propose a minimal biophysical mechanism based on the 'template mechanism' of vegetative cell growth assuming that glycans are oriented perpendicular to the long axis of the cell (*Figure 3A*) (*Koch and Doyle, 1985*; *Höltje, 1998*; *Domínguez-Escobar et al., 2011*; *Garner et al., 2011*; *Beeby et al., 2013*; *Dover et al., 2015*), without requiring any further assumptions about the outer cell wall structure of Gram-positive bacteria, which is still unclear (*Hayhurst et al., 2008*; *Beeby et al., 2013*; *Dover et al., 2015*). In this mechanism, a new glycan strand is inserted using both the septal glycan and leading forespore-proximal glycan strand of the lateral wall as template strands to which the new PG strand is cross linked. Subsequently, peptide cross-links between the two template strands are removed from the mother-cell proximal side by the DMP complex. Specifically, in this complex SpoIIP has well documented endopeptidase activity (*Morlot et al., 2010*). Note, similar 'make-before-break' mechanisms were proposed to allow vegetative cell wall growth without reducing cell wall integrity (*Koch and Doyle, 1985*; *Höltje, 1998*). A more detailed mechanism that requires the insertion of multiple new glycan strands to account for glycan removal by SpoIID is shown in *Figure 3—figure supplement 1*. In either model, synthesis of new PG at the LE likely occurs before degradation, thereby naturally preventing cell lysis during engulfment.

The coordination between PG insertion from the forespore and removal by DMP in the mother cell could lead to movement of the junction between the septal peptidoglycan and the lateral peptidoglycan around the forespore to mediate successful engulfment. Based on this proposed mechanism, we created a model whereby insertion and degradation happens, for simplicity, simultaneously by an insertion-degradation complex (IDC), also reflecting the high degree of coordination suggested by the template mechanism. In this model IDC recognizes the leading edge and inserts glycan polymers perpendicular to the long axis of the cell (*Figure 3B*). Additionally, the model proposes that IDC can recognize glycan ends and initiate glycan polymerization from the end defect with probability of repair $p_{rep}$. During glycan insertion, when an IDC encounters a gap in the outer cell wall strands, it continues polymerization with probability of processivity $p_{pro}$ (*Figure 3C*). A systematic exploration of the above model parameters showed that intact spores form for $p_{rep}$ and $p_{pro} > 0.8$ with a marginal dependence on the number of IDCs (*Figure 4G*, *Figure 4—figure supplement 1*). However, to compare the model with microscopy data we require a 3D dynamic implementation of this model that reflects the stochasticity of underlying molecular events.

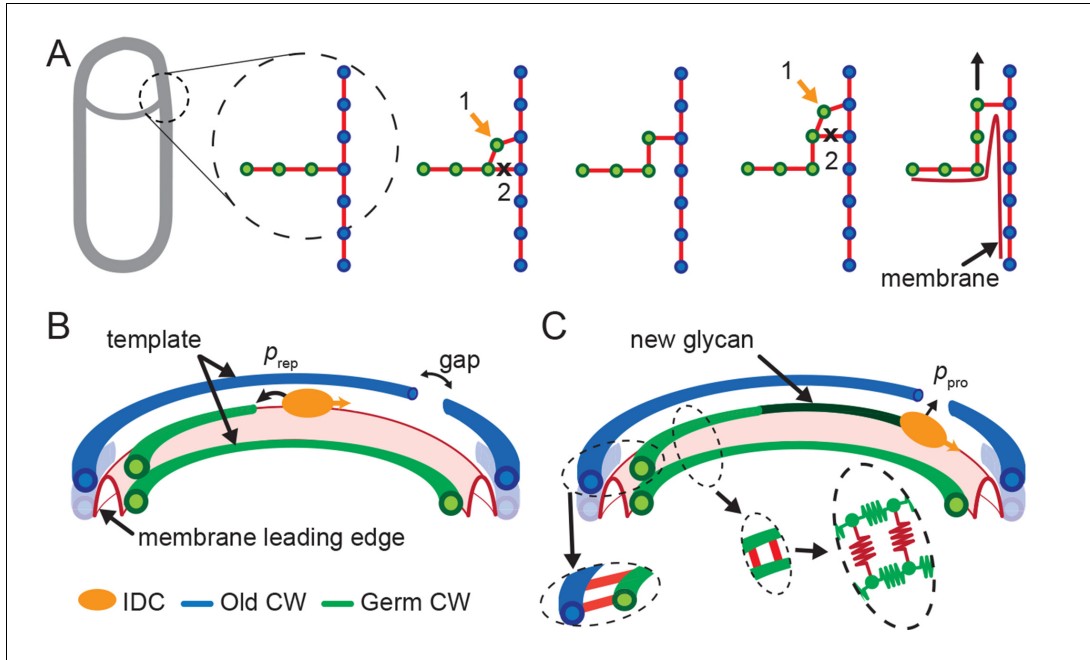

**Figure 3.** Template model for leading edge (LE) movement. (**A**) Cell cross-section with glycan strands in the plane perpendicular to the long axis of the cell. One strand from old cell wall (blue) and one strand from newly synthesized germ-cell wall (green) are used as a template for new glycan insertion. Coordination between glycan insertion (orange arrow) and peptide cross-link degradation (black cross) drives LE forward. (**B**) 3D model of stochastic glycan insertion by insertion-degradation complex (IDC) with transpeptidase and transglycosylase activity. Probability of IDC to start inserting new glycan from old glycan end and repair end defect is $p_{\mathrm{rep}}$. (**C**) New inserted glycan shown in dark green. Probability of IDC to continue glycan insertion when it encounters gap in old cell wall is probability of processivity $p_{\mathrm{pro}}$. (Inset) Horizontal (between old and new glycan strands) and vertical (between new glycan strands) peptide links are shown in red. In our coarse-grained model glycans are simulated as semi-flexible filaments consisting of beads (green) connected with springs (green). Peptides are simulated as springs (red) connecting neighboring glycan beads.

The following figure supplement is available for figure 3:

**Figure supplement 1.** Extended models that account for glycan-strand degradation.

## Langevin simulations reproduce observed phenotypes

To simulate stochastic insertion at the leading edge we used Langevin dynamics of a coarse-grained PG meshwork (see Materials and methods). Briefly, glycan strands are modeled as semi-flexible filaments consisting of beads connected with springs, while peptide bridges are modeled as springs connecting glycan beads (*Figure 3C*) (*Laporte et al., 2012*; *Tang et al., 2014*; *Huang et al., 2008*). Typical length of inserted glycan polymer is ~1 μm (~1/3 cell circumference) (*Hayhurst et al., 2008*) and in our model the peptide bridges between newly inserted glycan strands are in a relaxed state. Glycan beads experience forces due to glycan elastic springs ($k_{\mathrm{gly}}$), glycan persistence length ($l_{\mathrm{p}}$), elastic peptide links ($k_{\mathrm{pep}}$), stochastic thermal fluctuations, and pressure difference ($\Delta p$) between forespore and mother cell (see *Equation 1* and Appendix 2). Glycan strands in the PG layer are connected with neighboring glycans by stem peptides (*Figure 4A*). In our model, the angle between neighboring stem peptides that belong to the same glycan strand is assumed to be 90° (*Nguyen et al., 2015*; *Huang et al., 2008*). Therefore, every other stem peptide is in plane with the glycan sheet. In our model $\Delta p$ originates from the packing of the *B. subtilis* chromosome (~4.2 Mbp) in the small forespore compartment (*Errington, 1993*; *Perez et al., 2000*; *Bath et al., 2000*; *Yen Shin et al., 2015*).

To systematically explore the peptidoglycan parameters, we compared our simulations with actual changes in forespore volume, forespore surface area, and percentage of engulfment

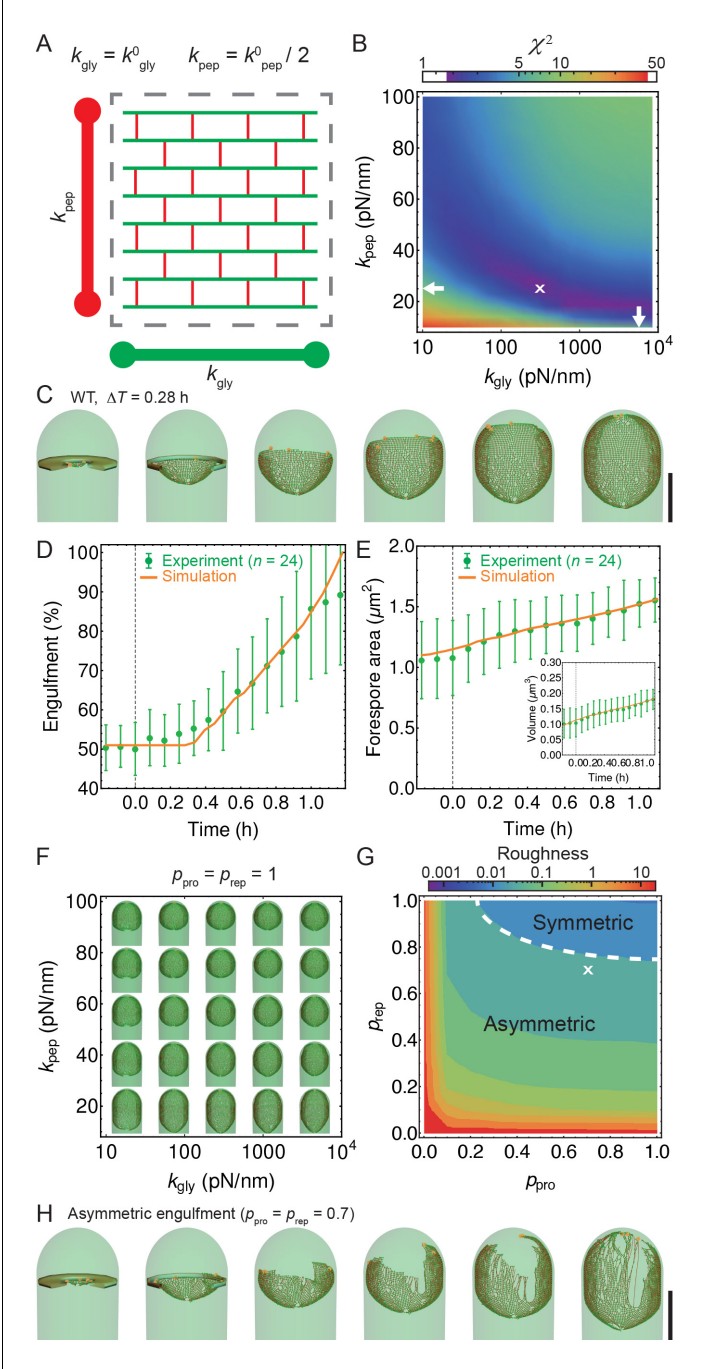

**Figure 4.** Template model reproduces experimentally observed phenotypes. (**A**) Effective spring constants in our model represent coarse-grained PG network. Here the angle between neighboring stem peptides that belong to a single glycan is assumed to be 90°. Therefore, every other stem peptide is in plane with glycan sheet (**Nguyen et al., 2015**, **Huang et al., 2008**). The role of effective glycan persistence length on engulfment is negligible (see **Figure 4—figure supplement 3**). (**B**) Simulations for different values of effective peptide $k_{pep}$ and glycan $k_{gly}$ spring constants are compared with experimentally measured forespore surface area, volume and engulfment using mutual $\chi^2$ statistics (**Equation 2**). Arrows point to effective literature $k_{pep}$ and $k_{gly}$ (**Nguyen et al., 2015**). Dark blue region corresponds to simulation parameters that best fit experimental data (**Figure 4—figure supplement 4**, **Video 3**). For large enough $k_{gly} > 200$ pN/nm mutual $\chi^2$ is almost independent of $k_{gly}$. (**C**) Snapshots of WT simulations for parameters ($k_{gly} = 200$ pN/nm, $k_{pep} = 25$ pN/nm, $N_{IDC} = 5$) marked with '×' in panel (**B**) (**Video 2**). The thick septum is treated as outer cell wall, and is assumed degraded once IDCs move along. (**D–E**) Time traces of experimentally measured engulfment, forespore surface area and forespore volume
*Figure 4 continued on next page*

*Figure 4 continued*

(green) in comparison with results from a single simulation (orange). Parameters used in simulation are marked with '×' in panel (**B**). For all other parameters see Appendix 2, Appendix-table 1. (**F**) Snapshots of fully engulfed forespores for various peptidoglycan elastic constants. (**G**) For various values of independent parameters $p_{\mathrm{rep}}$ and $p_{\mathrm{pro}}$ roughness of the LE is calculated at the end of stochastic simulations (see *Figure 4—figure supplement 1*, and *Video 4*). Here 0 roughness correspond to perfectly symmetric LE; for high enough $p_{\mathrm{rep}} = p_{\mathrm{pro}} > 0.8$ LE forms symmetric profiles. (**H**) Simulation for asymmetric engulfment is obtained for same parameter as WT except $p_{\mathrm{rep}} = p_{\mathrm{pro}} = 0.7$ (marked with '×' in panel (**G**)). Average ± SD. Scale bars 1 μm.

The following figure supplements are available for figure 4:

**Figure supplement 1.** Simulation of the stochastic model of insertion at the leading edge (LE).

**Figure supplement 2.** In simulations majority of peptide extensions are in the linear elastic regime.

**Figure supplement 3.** Engulfment is unaffected by glycan persistence length.

**Figure supplement 4.** Simulations with different peptidoglycan (PG) elastic constants.

**Figure supplement 5.** Simulations with decoupled synthesis and degradation.

extracted from time-lapse movies, using $\chi^2$ fitting (*Figure 4B*, *Equation 2*, Materials and methods). Parameters that best fit experimental measurements belong to dark blue region in agreement with molecular dynamic simulations (*Nguyen et al., 2015*). For a single peptide bond, the linear elasticity regime is valid for extensions that are less than 1 nm (*Nguyen et al., 2015*) and this elastic regime is maintained in the regions with low $\chi^2$ (*Figure 4—figure supplement 2*). For large enough glycan stiffness ($k_{\mathrm{gly}} > 300$ pN/nm) $\chi^2$ becomes independent of $k_{\mathrm{gly}}$ (*Figure 4B*). A typical simulation shown in *Figure 4C* matches experimental measurements of time-dependent engulfment, volume, and sur-face area (*Figure 4D,E*). PG spring constants drastically affect forespore morphologies. By decreasing $k_{\mathrm{pep}}$ forespores elongate, while by increasing $k_{\mathrm{pep}}$ forespores shrink, as measured along the long axis of the cell. Changing $k_{\mathrm{gly}}$ has only minor effects on volume and surface area. However, the main effect is on forespore curvature (see *Figure 4—figure supplement 4*): high $k_{\mathrm{gly}}$ increases the curvature of forespore ends (making them more pointy), while low $k_{\mathrm{gly}}$ decreases the curvature of the forespore ends. Our simulations successfully reproduce asymmetric engulfment (*Figure 4F,G*; *Video 5*). For $p_{\mathrm{rep}}$ and $p_{\mathrm{pro}} \lesssim 0.8$ we obtained asymmetric engulfment that reproduces the phenotypes observed when PG synthesis or degradation is partially blocked. When defects in the peptidoglycan meshwork are not repaired, different parts of the leading edge extend in an uncoordinated manner, producing asymmetric engulfment.

Since our simulations correctly reproduced engulfment dynamics we used simulation parameters to estimate glycan insertion velocities $V_{\mathrm{IDC}}$ of IDC (see Appendix 2). Using this method we estimated a lower bound on product $N_{\mathrm{IDC}} \cdot V_{\mathrm{IDC}} \sim 110$ nm/s, where $N_{\mathrm{IDC}}$ is the number of insertion complexes. Similarly, by estimating

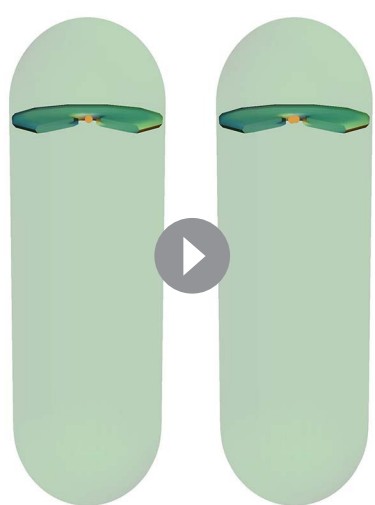

**Video 3.** Simulations of WT (left) and asymmetric engulfment (right). Parameters are the same ($k_{\mathrm{pep}} = 25$ pN/nm, $k_{\mathrm{gly}} = 200$ pN/nm, $N_{\mathrm{IDC}} = 5$) except for WT engulfment $p_{\mathrm{rep}} = p_{\mathrm{pro}} = 1$ and for asymmetric engulfment $p_{\mathrm{rep}} = p_{\mathrm{pro}} = 0.7$. For full exploration of stochastic insertion parameters see *Video 4* and *Figure 4—figure supplement 1*. Front opening of the forespore is not shown for clarity.

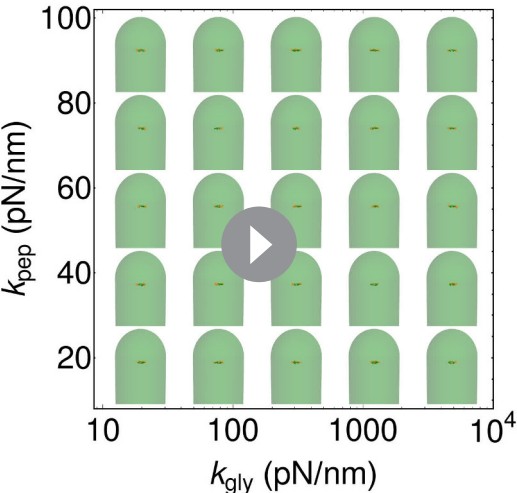
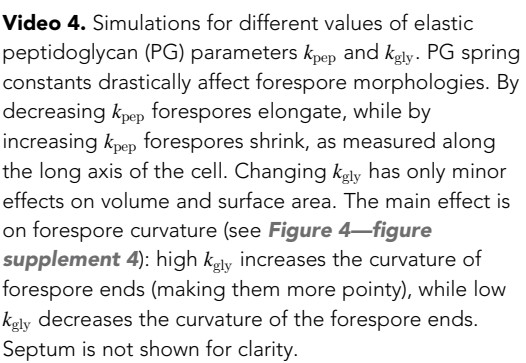

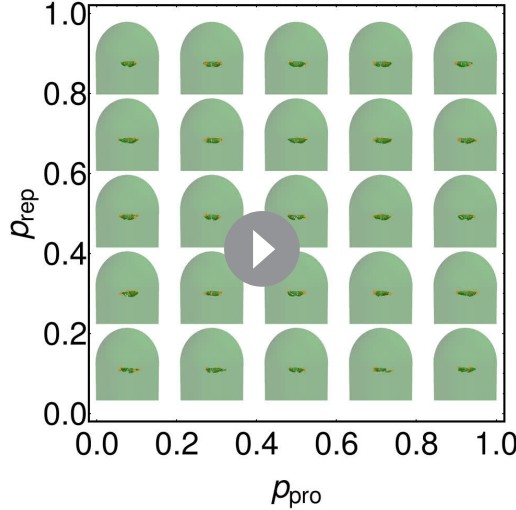

**Video 4.** Simulations for different values of elastic peptidoglycan (PG) parameters $k_{pep}$ and $k_{gly}$. PG spring constants drastically affect forespore morphologies. By decreasing $k_{pep}$ forespores elongate, while by increasing $k_{pep}$ forespores shrink, as measured along the long axis of the cell. Changing $k_{gly}$ has only minor effects on volume and surface area. The main effect is on forespore curvature (see *Figure 4—figure supplement 4*): high $k_{gly}$ increases the curvature of forespore ends (making them more pointy), while low $k_{gly}$ decreases the curvature of the forespore ends. Septum is not shown for clarity.

**Video 5.** Simulations for different values of stochastic parameters $p_{rep}$ and $p_{pro}$. Decreasing $p_{rep}$ and $p_{pro}$ below 0.8 results in asymmetric engulfment. For full exploration of stochastic insertion parameter see *Figure 4—figure supplement 1*.

the total amount of newly inserted material in the forespore within ~0.8 hr without any pausing we obtain $N_{IDC} \cdot V_{IDC} \sim 117$ nm/s. For circumferentially processive PBPs (PbpA and PbpH), the absolute velocity measured using TIRF microscopy is ~20–40 nm/s during vegetative cell growth (*Domínguez-Escobar et al., 2011*; *Garner et al., 2011*), which is in agreement with the speed of forespore GFP-MreB determined from our TIRF experiments ((28 ± 8) nm/s, $n$ = 14; *Figure 2E*). Using this estimate for $V_{IDC}$, we obtain a lower bound 3–6 on the number of active, highly processive PBP molecules. However, the actual number of proteins could be higher for other nonprocessive PBPs (*Domínguez-Escobar et al., 2011*; *Garner et al., 2011*).

## Discussion

The results presented here suggest that engulfment involves coordinated PG synthesis and degradation processes that are segregated between different cell types: first, PG is synthesized in front of the LE of the engulfing membrane by a forespore-associated PG biosynthetic machinery that rotates following the LE of the engulfing membrane. Then this new PG is targeted for degradation by the mother cell-associated PG degradation machinery comprised of the DMP complex (*Figure 2H*). The delocalization of DMP when PG synthesis is inhibited with antibiotics (*Figure 2*, *Figure 2—figure supplement 3*) indicates that the DMP either forms an actual complex with the PG biosynthetic machinery across the septal PG (to form a single insertion degradation complex (IDC), as shown in *Figure 3*) or that DMP targets the new PG synthesized at the LE of the engulfing membrane. In the latter, DMP might specifically target the cross-links that attach the old lateral cell wall to the new PG synthesized at the LE of the engulfing membrane (*Figure 2H*, orange). Since those cross-links join old, modified PG from the lateral cell wall to newly synthesized PG at the LE, those peptide bridges might have a unique chemical composition or structural arrangement that could be specifically recognized by DMP. Hence, either approach provides a safety mechanism during engulfment, since it would prevent DMP from degrading the old PG of the lateral cell wall, which could lead to cell lysis.

We have conceptualized these results in a biophysical model in which a PG insertion-degradation complex (IDC), representing PBPs for PG synthesis and DMP proteins for PG degradation, catalyzes PG remodeling at the LE of the engulfing membrane. Specifically, we propose that new glycan strands are inserted ahead of the LE of the engulfing membrane and PG is degraded on the mother cell proximal side to create space for forward movement of the LE (*Figure 3*). This is similar to the 'make-before-break' model of vegetative cell-wall growth, which postulates that the vegetative cell wall is elongated by inserting new PG strands prior to degrading old strands (*Koch and Doyle, 1985*) (although bacteria can also make a *de novo* cell wall (*Ranjit and Young, 2013*, *Kawai et al., 2014*). The make-before-break mechanism also accounts for the directional movement of the LE towards the forespore pole, since the substrate for DMP is new PG synthesized by forespore PBPs, which is always ahead of the LE of the engulfing membrane.

Using Langevin simulations we successfully reproduced the dynamics of engulfment, forespore volume, and surface area. Additionally, our model correctly reproduced asymmetric engulfment observed with reduced IDC activity, and we estimated that with only a handful of highly processive PBP molecules are necessary to reproduce the observed LE dynamics. A more general model without strong coupling between the PG biosynthetic and PG degradation machineries also leads to successful engulfment (Appendix 2, *Figure 4—figure supplement 5*, *Video 6*). However, DMP has to be guided to degrade only the peptide cross-links between old and new glycan strands, and should also prevent detachment of the septal peptidoglycan from the old cell wall.

Since our simple mechanism in *Figure 3A* entails hydrolysis of certain peptide bonds but no glycan degradation, we explored additional mechanisms since the SpoIID protein of the DMP complex shows transglycosylase activity (*Morlot et al., 2010*). First, it is possible that engulfment entails a two-for-one mechanism, with two new glycan strands are added and the newly inserted glycan strand at the LE is degraded (*Höltje, 1998*) (*Figure 3—figure supplement 1A*). Similarly, the three-for-one mechanism would also work (*Scheffers and Pinho, 2005*). Second, one new glycan strand might be added and the inner most cell-wall glycan of the thick, lateral cell wall degraded (*Figure 3—figure supplement 1B*). This would make the lateral cell wall thinner as the engulfing membrane moves forward (*Tocheva et al., 2013*). Finally, it is possible that insertion and degradation are not intimately coupled, and that there is simply a broad region in which PG is inserted ahead of the engulfing membrane, to create multiple links between the septal PG and the lateral cell wall (as shown in *Figure 2H*), and that the DMP complex has a preference for newly synthesized PG. All of these models require the spatial coordination between cell wall degradation and synthesis to avoid compromising cell wall integrity and inducing cell lysis, and all share a common 'make-before-break' strategy to promote robustness of the otherwise risky PG remodeling process (*Koch and Doyle, 1985*). In order to waste as little energy as possible, a more stringent 'make-just-before-break' strategy may even apply, motivating more intimate coupling between the PG biosynthetic and degradation machineries.

Our simple biophysical mechanism postulates that engulfment does not rely on pulling or pushing forces for membrane migration. Instead, cell wall remodeling makes room for the mother cell membrane to expand around the forespore by entropic forces. During engulfment the mother-cell surface area increases by ~2 μm$^2$ (~25%, see *Figure 1—figure supplement 3*), and this excess of membrane could simply be accommodated around the forespore by remodeling the PG at the LE. However, our model does not include all potential contributors to engulfment. For instance, the SpoIIQ-AH zipper, which is dispensable for engulfment in native conditions (*Broder and Pogliano, 2006*), might prevent membrane backward movement, and might also help localize the IDC components toward the LE. Interestingly, SpoIIQ-AH interaction is essential for engulfment in Clostridium difficile where the SpoIIQ ortholog posseses endopeptidase activity (*Crawshaw et al., 2014*; *Serrano et al., 2016*; *Fimlaid et al., 2015*). The model also

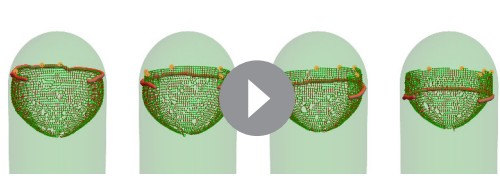

**Video 6.** Simulations with decoupled synthesis and degradation. New glycans are released from the old cell wall with typical delay time $\tau_{delay}$. Simulations for four different values of $\tau_{delay} = 0$, 0.9, 9, and 18 min (from left to right). For longer $\tau_{delay}$ the larger is separation between synthesis and membrane leading edge that is shown as red cylinder.

does not consider the impact of the tethering of the LE of the engulfing membrane to the forespore via interactions between the mother cell membrane anchored DMP complex at the LE and forespore synthesized PG. Future experiments and modeling should address the role of these and other potential contributors to LE migration, which will allow us to refine our biophysical model and obtain a comprehensive view of membrane dynamics during engulfment. Furthermore, understanding the cooperation between PBPs and DMP will provide valuable clues about the structure of the cell wall in Gram-positive bacteria.

## Materials and methods

### Strains and culture conditions

All the strains used in this study are derivatives of *B. subtilis* PY79. Complete lists of strains, plasmids, and oligonucleotides see Appendix 3. Detailed descriptions of plasmid construction are provided in *Supplementary file 1*. For each experiment we had at least two biological replicas, and each one contains at least three technical replicas. Averages of individual cells, but not the averages of different replicas are reported. Sporulation was induced by resuspension (*Sterlini and Mandelstam, 1969*), except that the bacteria were grown in 25% LB prior to resuspension, rather than CH medium. Cultures were grown at 37°C for batch culture experiments, and at 30°C for timelapse experiments.

### Fluorescence microscopy

Cells were visualized on an Applied Precision DV Elite optical sectioning microscope equipped with a Photometrics CoolSNAP-HQ$^2$ camera and deconvolved using SoftWoRx v5.5.1 (Applied Precision). When appropriate, membranes were stained with 0.5 µg/ml FM 4–64 (Life Technologies, Waltham, Massachusetts) or 1 µg/ml Mitotracker green (Life Technologies). Cells were transferred to 1.2% agarose pads for imaging. The median focal plane is shown.

### Timelapse fluorescent microscopy

Sporulation was induced at 30°C. 1.5 hr after sporulation induction, 0.5 µg/ml FM 4–64 was added to the culture and incubation continued for another 1.5 hr. Seven µl samples were taken 3 hr after resuspension and transferred to agarose pads prepared as follows: 2/3 vol of supernatant from the sporulation culture; 1/3 vol 3.6% agarose in fresh A+B sporulation medium; 0.17 µg/ml FM 4–64. When appropriated, cephalexin (50 µg/ml) or bacitracin (50 µg/ml) was added to the pad. Pads were partially dried, covered with a glass slide and sealed with petroleum jelly to avoid dehydration during timelapse imaging. Petroleum jelly is not toxic and cannot be metabolized by *B. subtilis*, which poses an advantage over other commonly used sealing compounds, such as glycerol, which can be used as a carbon source and inhibit the initiation of sporulation. Pictures were taken in an environmental chamber at 30°C every 5 min for 5 hr. Excitation/emission filters were TRITC/CY5. Excitation light transmission was set to 5% to minimize phototoxicity. Exposure time was 0.1 s.

### Forespore GFP-MreB tracking experiments

MreB tracking experiments were performed using the strain JLG2411, which produced GFP-MreB in the forespore after polar septation from *spoIIQ* promoter. Sporulation and agarose pads were done as described in Timelapse fluorescent microscopy, except that FM 4–64 was only added to the agarose pads and not to the sporulating cultures. A static membrane picture was taken at the beginning of the experiment, and was used as a reference to determine the position of the GFP-MreB foci. GFP-MreB motion at the cell surface was determined by TIRF microscopy (*Garner et al., 2011*; *Domínguez-Escobar et al., 2011*), taking pictures every 4 s for 100 s. Imaging was performed at 30°C. We used two different microscopes to perform TIRF microscopy: (i) An Applied Precision Spectris optical sectioning microscope system equipped with an Olympus IX70 microscope, a Photometrics CoolSNAP HQ digital camera and a 488 nm argon laser. To perform TIRF in this microscope, we used an Olympus 1003 1.65 Apo objective, immersion oil $n$ = 1.78 (Cargille Laboratories), and sapphire coverslips (Olympus). Laser power was set to 15%, and exposure time was 200 ms. (ii) An Applied Precision OMX Structured Illumination microscopy equipped with a Ring-TIRF system and a UApoN 1.49NA objective, immersion oil $n$ = 1.518. Exposure time was 150 ms.

Images were analyzed using the ImageJ-based FIJI package. Sporangia were aligned vertically using the rotation function in FIJI. GFP-MreB foci were tracked using the TrackMate pluging (**Tinevez et al., 2016**), using the LoG detector, estimated blob diameter of 300 nm, simple LAP tracked and linking max distance of 300 nm. Only tracks that contained more than four points were used to determine the MreB foci speed.

## Image analysis

We used the semi-automated active contour software JFilament available as ImageJ plugin to extract fluorescently labeled membrane position over time (**Smith et al., 2010**). Membrane position obtained from the medial focal plane is used in custom built Mathematica software to calculate 3D volume and surface area by assuming rotational symmetry around the axis connecting the center of masses of mother cell and forespore. For available code and example see **Supplementary file 2** . Kymographs as in **Figure 1E** were created by collecting intensities along the forespore contours. Subsequently, pixel angles were determined using pixel position relative to the mother-forespore frame as defined in inset of **Figure 1E**. Forespore fluorescent intensities along angles are normalized and interpolated using third-order polynomials. For a given angle the population intensity average of different cells is calculated and plotted over time. Time 0' is the onset of septum curving.

## Quantification of GFP-SpoIID, GFP-SpoIIM and GFP-SpoIIP fraction at LE

Antibiotics were added 2 hr after resuspension, and samples were taken one hour later for imaging. Exposure times and image adjustments were kept constant throughout the experiment. To determine the fraction of GFP signal at the LE, GFP pixel intensities of seven optical sections covering a total thickness of 0.9 μm were summed. GFP intensities at the LE ($I_{LE}$) and in the rest of the mother cell ($I_{MC}$) were determined separately by drawing polygons encompassing the LE or the MC. After subtraction of the average background intensity, the fraction of GFP fluorescence at LE ($\frac{I_{LE}}{I_{LE}+I_{MC}}$) was determined for each sporangium.

## Langevin dynamics

The Langevin dynamic equation of the $i^{th}$ bead at position $\mathbf{r}_i$ is given by:

$$\zeta_i \frac{d\mathbf{r}_i}{dt} = \mathbf{F}_i^{spr} + \mathbf{F}_i^{bend} + \mathbf{F}_i^{pep} + \mathbf{F}_i^{stoch} + \mathbf{F}_i^{\Delta p} + \mathbf{F}_i^{wall}, \tag{1}$$

where the left-hand side depends on the drag coefficient $\zeta_i \approx 4\pi\eta_{med}l_0$ (**Howard, 2001**), with $\eta_{med}$ is the medium viscosity and $l_0$ equilibrium distance between neighbouring beads (see Appendix 1). On the right-hand side of **Equation 5** we have contributions of glycan elastic spring, glycan bending, peptide elastic links, stochastic thermal fluctuations, pressure difference $\Delta p$ between forespore and mother, and excluded volume from the old cell wall, respectively.

## $\chi^2$ fitting of parameters

To compare simulations with experiments we measured forespore volume ($V_i$), forespore surface area ($S_i$) and engulfment ($E_i$) and constructed a quality-of-fit function as:

$$\chi^2 = \sum_i \left[ \frac{(V_i^{exp} - V_i^{sim})^2}{\sigma^2(V_i^{exp})} + \frac{(S_i^{exp} - S_i^{sim})^2}{\sigma^2(S_i^{exp})} + \frac{(E_i^{exp} - E_i^{sim})^2}{\sigma^2(E_i^{exp})} \right], \tag{2}$$

where index $i$ corresponds to $i^{th}$ time point, and $\sigma$ is the standard deviation (**Spitzer et al., 2006**).

## Acknowledgements

This work is supported by European Research Council Starting Grant 280492-PPHPI to NO and RGE, EMBO Long Term Fellowship to JLG, National Institutes of Health R01-GM57045 to KP. We thank the Microscopy Core at UC San Diego (P30 NS047101) for help in using the Applied Precision OMX Structured Illumination microscopy used for TIRF experiments. The funders had no role in study design, data collection and analysis, decision to publish, or preparation of the manuscript.

## Additional information

### Funding

| Funder | Grant reference number | Author |
|---|---|---|
| European Research Council | 280492-PPHPI | Nikola Ojkic<br>Robert G Endres |
| European Molecular Biology Organization | ATLF1274-2011 | Javier López-Garrido |
| National Institutes of Health | R01-GM57045 | Kit Pogliano |

The funders had no role in study design, data collection and interpretation, or the decision to submit the work for publication.

### Author contributions

NO, Modeling and simulations, Conception and design, Analysis and interpretation of data, Drafting or revising the article; JL-G, Conception and design, Acquisition of data, Analysis and interpretation of data, Drafting or revising the article; KP, RGE, Conception and design, Analysis and interpretation of data, Drafting or revising the article

### Author ORCIDs

Robert G Endres, http://orcid.org/0000-0003-1379-659X

## Additional files

### Supplementary files

• Supplementary file 1. Plasmid construction.

• Supplementary file 2. Image analysis example with code.

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

## Appendix 1

## Image analysis

### Forespore volume and surface area

Forespore volume and surface area are estimated from tracked fluorescent membranes in the medial focal plane using ImageJ plugin JFilament (*Smith et al., 2010*). JFilament is a semi-automated active contour software that is used for tracking fluorescently labelled membrane over time. The output of the software is a string of discrete membrane dots $\mathbf{r}_i = (x_i, y_i)$. A typical distance between neighbouring dots is $l_i \sim 1$ pixel. From the positions of the membrane dots, a costume-built Mathematica program was used to calculate the 3D volume ($V$) and surface area ($S$) by assuming rotational symmetry around the axis connecting the center of mass of the forespore and mother cell. The volume is given by:

$$V = \frac{1}{2} \sum_{i=1}^{N} \pi d_i^2 \, l_i |\hat{\mathbf{t}}_i \, \hat{\mathbf{e}}_{\mathrm{fm}}|, \tag{3}$$

where $N$ is the total number of dots, $d_i$ is the shortest distance between $i^{\mathrm{th}}$ dot and rotational axis, $l_i = \sqrt{(x_{i+1} - x_i)^2 + (y_{i+1} - y_i)^2}$ is the distance between neighbouring dots, and $\hat{\mathbf{t}}_i \equiv (\mathbf{r}_{i+1} - \mathbf{r}_i)/\mathbf{r}_{i+1} - \mathbf{r}_i$ is the unit tangent vector, and $\hat{\mathbf{e}}_{\mathrm{fm}}$ is the unit vector of the rotational axis. Since the sum extends over all the dots we used prefactor $\frac{1}{2}$ in order to correct for double counting. Similarly, the surface area is:

$$S = \frac{1}{2} \sum_{i=1}^{N} 2\pi d_i \, l_i. \tag{4}$$

## Calculating gap arc length

Forespore membrane contours are extracted as described in Forespore volume and surface area. Using a simple thresholding method (0.55 ± 0.05, relative to bright engulfing cup) the part of forespores that is not covered with mother membrane is selected. The total arc length is subsequently calculated for segments not covered with the mother membrane. Analysis of cells with symmetric and asymmetric cups are included in the analysis of the main text (*Figure 1H*).

## Appendix 2

### Model and simulations

#### Stochastic leading-edge insertion

In our model insertion-degradation complexes (IDC) drive leading-edge (LE) advancement. Glycan-strand insertion occurs exclusively at the leading edge (*Figure 3*). A single IDC binds to a previously created glycan defect with probability $p_{\text{rep}}$ (probability to repair) or anywhere along the LE with probability (1-$p_{\text{rep}}$). Once bound, the IDC inserts a glycan strand of a typical length 1 $\mu$m (*Hayhurst et al., 2008*). In the model IDC uses two glycan strands for guiding the insertion as suggested by the proposed template model of vegetative cell growth (*Höltje, 1998*). One template strand belongs to the elongating septal PG and other strand to the old cell wall. During the insertion process, if IDC encounters a gap in the old cell wall, IDC continues insertion with probability $p_{\text{pro}}$ (processivity probability) or terminates insertion with probability (1-$p_{\text{pro}}$). When the IDC reaches the end of the germ cell wall template, insertion is terminated.

To explore general properties of above simple stochastic model we discretized glycan strands in segments of 2 nm, which corresponds to a distance between two neighboring antiparallel peptide bonds (*Figure 4A*, *Figure 4—figure supplement 1*). We simulated this simple model assuming that the total number $N_{\text{IDC}}$ of IDCs is constant. Also, IDC inserts one glycan segment per time step. Simulations are run until the LE reaches the height of 1 $\mu$m (500 glycans). For simulated LE profiles we measured their width ($2\sqrt{\langle h_i^2 \rangle - \langle h_i \rangle^2}$) and roughness (1 - $C/C_0$), where $\langle...\rangle$ is the average over LE segments, $h_i$ the height of the $i^{\text{th}}$ LE segment, $C$ the LE circumference, and $C_0$ the cell circumference.

#### Langevin dynamics

Inserted glycans are equilibriated using Langevin dynamics in 3D (*Laporte et al., 2012*; *Tang et al., 2014*; *Ojkic et al., 2014*). The Langevin dynamic equation of the $i^{\text{th}}$ glycan bead at position $\mathbf{r}_i$ is given by:

$$\zeta_i \frac{\mathrm{d}\mathbf{r}_i}{\mathrm{d}t} = \mathbf{F}_i^{\text{spr}} + \mathbf{F}_i^{\text{bend}} + \mathbf{F}_i^{\text{pep}} + \mathbf{F}_i^{\text{stoch}} + \mathbf{F}_i^{\Delta \text{p}} + \mathbf{F}_i^{\text{wall}}, \tag{5}$$

where the left-hand side depends on the drag coefficient $\zeta_i \approx 4\pi\eta_{\text{med}}l_0$ (*Howard, 2001*), with $\eta_{\text{med}}$ is the medium viscosity and $l_0$ equilibrium distance between neighboring beads. On the right-hand side of *Equation 5* we have contributions from glycan elasticity, glycan bending, peptide elasticity, thermal fluctuations, pressure difference $\Delta p$ between forespore and mother cell, and excluded volume from the old cell wall, respectively. Simulation parameters are in *Appendix 2—table 1*. Below we describe each force contribution.

#### Glycan elastic force

The elastic force on the $i^{\text{th}}$ bead due to neighboring linear springs is given by:

$$\mathbf{F}_i^{\text{spr}} = -\frac{\partial E^{\text{spr}}}{\partial \mathbf{r}_i} = -\frac{k_{\text{gly}}}{2}\sum_{j=1}^{N-1}\frac{\partial(|\mathbf{r}_{j+1} - \mathbf{r}_j| - l_0)^2}{\partial \mathbf{r}_i}, \tag{6}$$

where $N$ is the total number of beads in the glycan.

#### Glycan bending

The bending force is given by

$$\mathbf{F}_i^{\text{bend}} = -\frac{\partial E^{\text{bend}}}{\partial \mathbf{r}_i} = \frac{\kappa_{\text{b}}}{l_0} \sum_{j=2}^{N-1} \frac{\partial(\hat{\mathbf{t}}_j \hat{\mathbf{t}}_{j-1})}{\partial \mathbf{r}_i}, \tag{7}$$

with $\hat{\mathbf{t}}_i \equiv (\mathbf{r}_{i+1} - \mathbf{r}_i)/\mathbf{r}_{i+1} - \mathbf{r}_i$ is the unit tangent vector, and $\kappa_{\text{b}}$ is the glycan flexural rigidity. We further simplified **Equation 7** using identity (**Pasquali and Morse, 2002**)

$$\frac{\partial \hat{\mathbf{t}}_i}{\partial \mathbf{r}_j} = \frac{1}{l_0}(\delta_{i+1,j} - \delta_{i,j})(\hat{\mathbf{I}} - \hat{\mathbf{t}}_i \hat{\mathbf{t}}_i^{\text{T}}), \tag{8}$$

where $\delta_{i,j}$ the Kroneker symbol, $\hat{\mathbf{I}}$ the unit matrix, and

$$\hat{\mathbf{t}}_i \hat{\mathbf{t}}_i^{\text{T}} \equiv \begin{pmatrix} t_{i,x}^2 & t_{i,x}t_{i,y} & t_{i,x}t_{i,z} \\ t_{i,x}t_{i,y} & t_{i,y}^2 & t_{i,y}t_{i,z} \\ t_{i,x}t_{i,z} & t_{i,y}t_{i,z} & t_{i,z}^2 \end{pmatrix}, \tag{9}$$

## Peptide elastic force

The force on the $i^{\text{th}}$ glycan bead due to peptide connections is:

$$\mathbf{F}_i^{\text{pep}} = -\frac{\partial E^{\text{pep}}}{\partial \mathbf{r}_i} = -\frac{k_{\text{pep}}}{2} \sum_j \frac{\partial(|\mathbf{r}_j - \mathbf{r}_i| - l_{0\text{p}})^2}{\partial \mathbf{r}_i}, \tag{10}$$

where the sum is over beads of neighboring glycans that have peptide connections with the $i^{\text{th}}$ bead. Here $l_{0\text{p}}$ is the equilibrium peptide length, and $k_{\text{pep}}$ is the peptide spring constant.

## Stochastic force

The stochastic force due to thermal noise is given by (**Pasquali and Morse, 2002**)

$$\langle \mathbf{F}_i^{\text{stoch}} \mathbf{F}_i^{\text{stoch T}} \rangle = \frac{2k_{\text{B}}T\zeta_i}{\Delta t}\hat{\mathbf{I}}, \tag{11}$$

with $k_{\text{B}}T$ the thermal energy and $\Delta t$ the simulation time step.

## Pressure force

In our model pressure difference ($\Delta p$) is due to translocated DNA

$$\mathbf{F}_i^{\Delta \text{p}} = \Delta S_i \Delta p\, \hat{\mathbf{n}}_i \tag{12}$$

with $\Delta S_i$ the surface segment corresponding to the $i^{\text{th}}$ bead, and $\hat{\mathbf{n}}_i$ is the unit normal vector. Parameter $\Delta p$ is estimated using the contact-value theorem of confined polymers in a thermal equilibrium (**Li et al., 2008**). The osmotic pressure in the forespore compartment due to translocated DNA is $p_{\text{f}} = \left(\frac{R_{\text{f}} - \sigma/2}{R_{\text{f}}}\right)^2 c\, k_{\text{B}}T$, where $R_{\text{f}}$ is the forespore radius, $\sigma$ is the DNA cross-section diameter, and $c$ is the number density of DNA at the forespore inner surface. For simplicity, we assumed that DNA density is constant throughout the forespore. Since $\sigma \ll R_{\text{f}}$ we neglected the numerical prefactor in the expression for osmotic pressure. Using the same expression for the osmotic pressure in mother-cell compartment and $V_{\text{m}}/V_{\text{f}} \sim 5$ at the end of engulfment (**Figure 4**, **Figure 1—figure supplement 3**) we estimated a lower bound for the osmotic pressure difference $\Delta p \sim 86.31$ kPa.

### Excluded volume

Excluded volume force from the lateral old cell wall was added to each glycan bead when the bead is within $l_0$ of the lateral cell wall. The magnitude of excluded volume force was 70 pN in the normal and inward direction of the lateral wall.

## Simulations with decoupled synthesis and degradation

To explore the possibility that synthesis and degradation are not tightly coupled as in our IDC model, we simulated delayed degradation of peptide bonds connecting lateral cell wall and newly synthesized glycan strands. For this purpose, we introduced a typical delay time $\tau_{\text{delay}}$ of peptide degradation in our simulation (**Figure 4—figure supplement 5**, **Video 6**). As expected, the spatial insertion-degradation separation increases with $\tau_{\text{delay}}$ (**Figure 4—figure supplement 5A,B**). As long as no errors are made, this mechanism also leads to successful forespore engulfment.

To investigate the role of errors in cutting peptide bonds we simulated the possibility that PG degradation also cuts neighbouring peptide bond (peptide connection in different planes) of newly synthesized germ cell wall with probability $p_{\text{pcut}}$ (**Figure 4—figure supplement 5C–D**). For relatively small $p_{\text{pcut}} = 0.1$, an irregular peptidoglycan meshwork is formed. As long as $p_{\text{pcut}}$ is small, intact forespores are formed.

We also simulated dislocalized DMP degradation upon antibiotic treatment when synthesis is stalled (**Figure 2E–F**, **Figure 2—figure supplement 3**). We explored the possibility that dislocated DMP randomly cuts old germ cell wall peptides with constant degradation rate $p_{\text{rpep}}$. In this scenario, irregular peptidoglycan networks protrude towards the mother cell with apparent volume increase while the leading edge remains still (**Figure 4—figure supplement 5E–F**). Similar phenotypes are experimentally observed about 2 hr after antibiotic treatment (see **Video 1**; **Figure 1—figure supplement 1A**)

## Numerical integration

After stochastic glycan insertion, **Equation 5** was numerically integrated with time step $\Delta t = 2 \cdot 10^{-8}$ s. The peptidoglycan (PG) network was equilibrated with 15,000 integration time steps. Simulations were also tested with 30,000 time steps to make sure that forespore volume, surface area, and engulfment remained unchanged. Obtained time traces of volume, surface area, and engulfment are subsequently rescaled in time to match experimental measurements (**Figure 4D–E**). A typical rescaling factor was $\widetilde{\Delta t} = 1.8 \cdot 10^5$. Since rescaling was done on fully equilibrated PG meshworks obtained relaxation dynamics were not affected by our rescaling method. From the mass conservation of inserted glycans we estimated $N_{\text{IDC}} \cdot V_{\text{IDC}} \sim N_{\text{in}} \, l_0 \, w / \widetilde{\Delta t}$, where $N_{\text{IDC}}$ is the number of IDC, $V_{\text{IDC}}$ is the IDC insertion velocity, $N_{\text{in}}$ is the number of inserted segments, $w = 7$ is the number of glycans per coarse-grained glycan (**Figure 4A**).

## Simulation parameters

**Appendix 2—table 1.** Model parameters.

| Symbol | Physical quantity | Values used in simulation | Sources / References | Notes |
|---|---|---|---|---|
| $T_0$ | Room temperature | 300 K | | |

*Appendix 2—table 1 continued on next page*

*Appendix 2—table 1 continued*

| Symbol | Physical quantity | Values used in simulation | Sources / References | Notes |
|---|---|---|---|---|
| $k_{pep}$ | Peptide effective spring constant; $k_{pep} = k_{pep}^0 / 2$ | 25 pN/nm | *Figure 4—figure supplement 4*, (*Nguyen et al., 2015*) | $k_{pep}^0$ for a single peptide |
| $k_{gly}$ | Glycan effective spring constant; $k_{gly} = k_{gly}^0$ | 5570 pN/nm | *Figure 4—figure supplement 4*, (*Nguyen et al., 2015*) | $k_{gly}^0$ for a single glycan |
| $l_{p0}$ | Glycan persistence length | 40 nm | *Figure 4—figure supplement 2*,(*Nguyen et al., 2015*) | |
| $\Delta p$ | Pressure difference | 86.31 kPa | Apendix (2.2) | |
| $\eta_{wat}$ | Water viscosity | 0.001 Pa s | | |
| $\eta_{med}$ | Medium viscosity | 1 Pa s | (*Spitzer et al., 2006*) | |
| $l_0 = l_{0p}$ | Mesh size | 0.014 $\mu m$ | | Our simulations |
| $\Delta t$ | Time step | $2 \cdot 10^{-8}$s | | Our simulations |

## Appendix 3

# Strains, plasmids and oligonucleotides

**Appendix 3—table 1.** Strains used in this study.

| Strain | Genotype or description | Reference, source or construction* |
|--------|------------------------|-----------------------------------|
| PY79 | Wild type | (*Youngman et al., 1984*) |
| ABS49 | ΔspoIIP::TetΩPspoIIP-GFP-spoIIPΩerm | (*Chastanet and Losick, 2007*) |
| ABS98 | ΔspoIIM::spcΩPspoIIM-GFP-spoIIDΩerm | (*Chastanet and Losick, 2007*) |
| ABS325 | ΔspoIID::kanΩPspoIID-GFP-spoIIDΩerm | (*Chastanet and Losick, 2007*) |
| JLG626 | ΔspoIIQ::erm | pJLG78 → PY79 (Em [R]) |
| JLG1420 | amyE::PspoIIQ-sfGFP-pbpFΩcat | pJLG213 → PY79 (Cm [R]) |
| JLG1421 | amyE::PspoIIR-sfGFP-pbpFΩcat | pJLG214 → PY79 (Cm [R]) |
| JLG1422 | thrC::PspoIID-sfGFP-pbpFΩspc | pJLG215 → PY79 (Sp [R]) |
| JLG1425 | amyE::PspoIIQ-sfGFP-pbpGΩcat | pJLG218 → PY79 (Cm [R]) |
| JLG1427 | amyE::PspoIIR-sfGFP-pbpGΩcat | pJLG219 → PY79 (Cm [R]) |
| JLG1428 | thrC::PspoIID-sfGFP-pbpGΩspc | pJLG220 → PY79 (Sp [R]) |
| JLG1555 | amyE::PspoIIQ-sfGFP-ponAΩcat | pJLG222 → PY79 (Cm [R]) |
| JLG1556 | amyE::PspoIIR-sfGFP-ponAΩcat | pJLG223 → PY79 (Cm [R]) |
| JLG1557 | thrC::PspoIID-sfGFP-ponAΩspc | pJLG230 → PY79 (Sp [R]) |
| JLG1558 | amyE::PspoIIQ-sfGFP-pbpDΩcat | pJLG224 → PY79 (Cm [R]) |
| JLG1559 | amyE::PspoIIR-sfGFP-pbpDΩcat | pJLG225 → PY79 (Cm [R]) |
| JLG1560 | thrC::PspoIID-sfGFP-pbpDΩspc | pJLG226 → PY79 (Sp [R]) |
| JLG1824 | amyE::PspoIIQ-sfGFP-pbpBΩcat | pJLG263 → PY79 (Cm [R]) |
| JLG1825 | amyE::PspoIIR-sfGFP-pbpBΩcat | pJLG264 → PY79 (Cm [R]) |
| JLG1826 | thrC::PspoIID-sfGFP-pbpBΩspc | pJLG265 → PY79 (Sp [R]) |
| JLG1827 | amyE::PspoIIQ-sfGFP-pbpHΩcat | pJLG266 → PY79 (Cm [R]) |
| JLG1828 | amyE::PspoIIR-sfGFP-pbpHΩcat | pJLG267 → PY79 (Cm [R]) |
| JLG1829 | thrC::PspoIID-sfGFP-pbpHΩspc | pJLG268 → PY79 (Sp [R]) |
| JLG1830 | amyE::PspoIIR-sfGFP-pbpIΩcat | pJLG270 → PY79 (Cm [R]) |
| JLG1831 | thrC::PspoIID-sfGFP-pbpIΩspc | pJLG271 → PY79 (Sp [R]) |
| JLG1832 | amyE::PspoIIQ-sfGFP-pbpAΩcat | pJLG272 → PY79 (Cm [R]) |
| JLG1833 | amyE::PspoIIR-sfGFP-pbpAΩcat | pJLG273 → PY79 (Cm [R]) |
| JLG1834 | thrC::PspoIID-sfGFP-pbpAΩspc | pJLG274 → PY79 (Sp [R]) |
| JLG1835 | amyE::PspoIIQ-sfGFP-pbpXΩcat | pJLG275 → PY79 (Cm [R]) |
| JLG1836 | amyE::PspoIIR-sfGFP-pbpXΩcat | pJLG276 → PY79 (Cm [R]) |
| JLG1837 | thrC::PspoIID-sfGFP-pbpXΩspc | pJLG277 → PY79 (Sp [R]) |
| JLG1838 | amyE::PspoIIQ-sfGFP-dacAΩcat | pJLG278 → PY79 (Cm [R]) |
| JLG1839 | amyE::PspoIIR-sfGFP-dacAΩcat | pJLG279 → PY79 (Cm [R]) |
| JLG1840 | thrC::PspoIID-sfGFP-dacAΩspc | pJLG280 → PY79 (Sp [R]) |
| JLG1851 | amyE::PspoIIQ-sfGFP-dacBΩcat | pJLG281 → PY79 (Cm [R]) |
| JLG1852 | amyE::PspoIIR-sfGFP-dacBΩcat | pJLG282 → PY79 (Cm [R]) |
| JLG1853 | thrC::PspoIID-sfGFP-dacBΩspc | pJLG283 → PY79 (Sp [R]) |
| JLG1854 | amyE::PspoIIQ-sfGFP-dacCΩcat | pJLG284 → PY79 (Cm [R]) |
| JLG1855 | amyE::PspoIIR-sfGFP-dacCΩcat | pJLG285 → PY79 (Cm [R]) |

*Appendix 3—table 1 continued on next page*

*Appendix 3—table 1 continued*

| Strain | Genotype or description | Reference, source or construction* |
|---|---|---|
| JLG1856 | thrC::PspoIID-sfGFP-dacCΩspc | pJLG286 → PY79 (Sp [R]) |
| JLG1857 | amyE::PspoIIQ-sfGFP-dacFΩcat | pJLG287 → PY79 (Cm [R]) |
| JLG1858 | thrC::PspoIID-sfGFP-dacFΩspc | pJLG289 → PY79 (Sp [R]) |
| JLG1859 | amyE::PspoIIQ-sfGFP-pbpIΩcat | pJLG269 → PY79 (Cm [R]) |
| JLG1860 | amyE::PspoIIR-sfGFP-dacFΩcat | pJLG288 → PY79 (Cm [R]) |
| JLG1861 | amyE::PspoIIQ-sfGFP-pbpEΩcat | pJLG296 → PY79 (Cm [R]) |
| JLG1863 | amyE::PspoIIR-sfGFP-pbpEΩcat | pJLG298 → PY79 (Cm [R]) |
| JLG1864 | thrC::PspoIID-sfGFP-pbpEΩspc | pJLG299 → PY79 (Sp [R]) |
| JLG2248 | amyE::PspoIIR-sfGFP-ponAΩcat ΔspoIIQ::erm | JLG626 → JLG1556 (Em [R]) |
| JLG2356 | ΔgerM::kan | pJLG361 → PY79 (Km [R]) |
| JLG2359 | amyE::PspoIIR-sfGFP-pbpAΩcat ΔspoIIQ::erm | JLG626 → JLG1833 (Em [R]) |
| JLG2360 | amyE::PspoIIR-sfGFP-pbpAΩcat ΔspoIIB::erm | KP343 → JLG1833 (Em [R]) |
| JLG2366 | amyE::PspoIIR-sfGFP-ponAΩcat ΔspoIIB::erm | KP343 → JLG1556 (Em [R]) |
| JLG2367 | amyE::PspoIIR-sfGFP-ponAΩcat ΔgerM::kan | JLG2356 → JLG1556 (Km [R]) |
| JLG2368 | amyE::PspoIIR-sfGFP-ponAΩcat ΔspoIIIAG-AH::kan | KP896 → JLG1556 (Km [R]) |
| JLG2369 | amyE::PspoIIR-sfGFP-ponAΩcat ΔspoIVFAB::cat::tet | KP1013 → JLG1556 (Tet [R]) |
| JLG2370 | amyE::PspoIIR-sfGFP-ponAΩcat ΔsigE::erm | KP161 → JLG1556 (Em [R]) |
| JLG2371 | amyE::PspoIIR-sfGFP-ponAΩcat spoIID::Tn917Ωerm | KP8 → JLG1556 (Em [R]) |
| JLG2372 | amyE::PspoIIR-sfGFP-ponAΩcat ΔspoIIP::tet | KP513 → JLG1556 (Tet [R]) |
| JLG2373 | amyE::PspoIIR-sfGFP-ponAΩcat spoIIM::Tn917Ωerm | KP519 → JLG1556 (Em [R]) |
| JLG2374 | amyE::PspoIIR-sfGFP-pbpAΩcat ΔgerM::kan | JLG2356 → JLG1833 (Km [R]) |
| JLG2375 | amyE::PspoIIR-sfGFP-pbpAΩcat ΔspoIIIAG-AH::kan | KP896 → JLG1833 (Km [R]) |
| JLG2376 | amyE::PspoIIR-sfGFP-pbpAΩcat ΔspoIVFAB::cat::tet | KP1013 → JLG1833 (Tet [R]) |
| JLG2377 | amyE::PspoIIR-sfGFP-pbpAΩcat ΔsigE::erm | KP161 → JLG1833 (Em [R]) |
| JLG2378 | amyE::PspoIIR-sfGFP-pbpAΩcat spoIID::Tn917Ωerm | KP8 → JLG1833 (Em [R]) |
| JLG2379 | amyE::PspoIIR-sfGFP-pbpAΩcat ΔspoIIP::tet | KP513 → JLG1833 (Tet [R]) |
| JLG2380 | amyE::PspoIIR-sfGFP-pbpAΩcat spoIIM::Tn917Ωerm | KP519 → JLG1833 (Em [R]) |
| JLG2411 | amyE::PspoIIQ-sfGFP-mreBΩcat | pJLG363 → PY79 (Cm [R]) |
| JLG2412 | amyE::PspoIIR-sfGFP-mreBΩcat | pJLG364 → PY79 (Cm [R]) |
| JLG2413 | thrC::PspoIID-sfGFP-mreBΩspc | pJLG365 → PY79 (Sp [R]) |
| JLG2414 | amyE::PspoIIQ-sfGFP-mblΩcat | pJLG371 → PY79 (Cm [R]) |
| JLG2415 | amyE::PspoIIR-sfGFP-mblΩcat | pJLG366 → PY79 (Cm [R]) |
| JLG2416 | thrC::PspoIID-sfGFP-mblΩspc | pJLG367 → PY79 (Sp [R]) |
| JLG2417 | amyE::PspoIIQ-sfGFP-mreBHΩcat | pJLG368 → PY79 (Cm [R]) |
| JLG2418 | amyE::PspoIIR-sfGFP-mreBHΩcat | pJLG369 → PY79 (Cm [R]) |
| JLG2419 | thrC::PspoIID-sfGFP-mreBHΩspc | pJLG370 → PY79 (Sp [R]) |
| KP8 | spoIID::Tn917Ωerm | (*Sandman et al., 1987*) |
| KP161 | ΔsigE::erm | (*Kenney and Moran, 1987*) |
| KP343 | ΔspoIIB::erm | (*Margolis et al., 1993*) |
| KP513 | ΔspoIIP::tet | (*Frandsen and Stragier, 1995*) |
| KP519 | spoIIM::Tn917Ωerm | (*Sandman et al., 1987*) |
| KP896 | ΔspoIIIAG-AH::kan | (*Blaylock et al., 2004*) |
| KP1013 | ΔspoIVFAB::cat::tet | (*Aung et al., 2007*) |

*Plasmid or genomic DNA employed (right side the arrow) to transform an existing strain (left side the arrow) into a new strain are listed. The drug resistance is noted in parentheses.

**Appendix 3—table 2.** Plasmids used in this study.

| Plasmid | Description |
| --- | --- |
| pJLG78 | ΔspoIIQ::erm |
| pJLG88 | amyE::PspoIIQ-pbpFΩcat |
| pJLG89 | amyE::PspoIIR-pbpFΩcat |
| pJLG90 | thrC::PspoIID-pbpFΩspc |
| pJLG91 | amyE::PspoIIQ-pbpGΩcat |
| pJLG92 | amyE::PspoIIR-pbpGΩcat |
| pJLG93 | thrC::PspoIID-pbpGΩspc |
| pJLG213 | amyE::PspoIIQ-sfGFP-pbpFΩcat |
| pJLG214 | amyE::PspoIIR-sfGFP-pbpFΩcat |
| pJLG215 | thrC::PspoIID-sfGFP-pbpFΩspc |
| pJLG218 | amyE::PspoIIQ-sfGFP-pbpGΩcat |
| pJLG219 | amyE::PspoIIR-sfGFP-pbpGΩcat |
| pJLG220 | thrC::PspoIID-sfGFP-pbpGΩspc |
| pJLG222 | amyE::PspoIIQ-sfGFP-ponAΩcat |
| pJLG223 | amyE::PspoIIR-sfGFP-ponAΩcat |
| pJLG224 | amyE::PspoIIQ-sfGFP-pbpDΩcat |
| pJLG225 | amyE::PspoIIR-sfGFP-pbpDΩcat |
| pJLG226 | thrC::PspoIID-sfGFP-pbpDΩspc |
| pJLG230 | amyE::PspoIIR-sfGFP-ponAΩcat |
| pJLG263 | amyE::PspoIIQ-sfGFP-pbpBΩcat |
| pJLG264 | amyE::PspoIIR-sfGFP-pbpBΩcat |
| pJLG265 | thrC::PspoIID-sfGFP-pbpBΩspc |
| pJLG266 | amyE::PspoIIQ-sfGFP-pbpHΩcat |
| pJLG267 | amyE::PspoIIR-sfGFP-pbpHΩcat |
| pJLG268 | thrC::PspoIID-sfGFP-pbpHΩspc |
| pJLG269 | amyE::PspoIIQ-sfGFP-pbpIΩcat |
| pJLG270 | amyE::PspoIIR-sfGFP-pbpIΩcat |
| pJLG271 | thrC::PspoIID-sfGFP-pbpIΩspc |
| pJLG272 | amyE::PspoIIQ-sfGFP-pbpAΩcat |
| pJLG273 | amyE::PspoIIR-sfGFP-pbpAΩcat |
| pJLG274 | thrC::PspoIID-sfGFP-pbpAΩspc |
| pJLG275 | amyE::PspoIIQ-sfGFP-pbpXΩcat |
| pJLG276 | amyE::PspoIIR-sfGFP-pbpXΩcat |
| pJLG277 | thrC::PspoIID-sfGFP-pbpXΩspc |
| pJLG278 | amyE::PspoIIQ-sfGFP-dacAΩcat |
| pJLG279 | amyE::PspoIIR-sfGFP-dacAΩcat |
| pJLG280 | thrC::PspoIID-sfGFP-dacAΩspc |
| pJLG281 | amyE::PspoIIQ-sfGFP-dacBΩcat |
| pJLG282 | amyE::PspoIIR-sfGFP-dacBΩcat |
| pJLG283 | thrC::PspoIID-sfGFP-dacBΩspc |

*Appendix 3—table 2 continued on next page*

*Appendix 3—table 2 continued*

| Plasmid | Description |
|---|---|
| pJLG284 | amyE::PspoIIQ-sfGFP-dacCΩcat |
| pJLG285 | amyE::PspoIIR-sfGFP-dacCΩcat |
| pJLG286 | thrC::PspoIID-sfGFP-dacCΩspc |
| pJLG287 | amyE::PspoIIQ-sfGFP-dacFΩcat |
| pJLG288 | amyE::PspoIIR-sfGFP-dacFΩcat |
| pJLG289 | thrC::PspoIID-sfGFP-dacFΩspc |
| pJLG296 | amyE::PspoIIQ-sfGFP-pbpEΩcat |
| pJLG298 | amyE::PspoIIR-sfGFP-pbpEΩcat |
| pJLG299 | thrC::PspoIID-sfGFP-pbpEΩspc |
| pJLG361 | ΔgerM::kan |
| pJLG363 | amyE::PspoIIQ-sfGFP-mreBΩcat |
| pJLG364 | amyE::PspoIIR-sfGFP-mreBΩcat |
| pJLG365 | thrC::PspoIID-sfGFP-mreBΩspc |
| pJLG366 | amyE::PspoIIR-sfGFP-mblΩcat |
| pJLG367 | thrC::PspoIID-sfGFP-mblΩspc |
| pJLG368 | amyE::PspoIIQ-sfGFP-mreBHΩcat |
| pJLG369 | amyE::PspoIIR-sfGFP-mreBHΩcat |
| pJLG370 | thrC::PspoIID-sfGFP-mreBHΩspc |
| pJLG371 | amyE::PspoIIQ-sfGFP-mblΩcat |

**Appendix 3—table 3.** Oligonucleotides used in this sudy.

| Primer | Sequence[†] |
|---|---|
| JLG-95 | CATGGATTACGCGTTAACCC |
| JLG-96 | GCACTTTTCGGGGAAATGTG |
| JLG-249 | catacgccgagttatcacatGATGATTCAACTGACAAATCTGG |
| JLG-250 | cacatttccccgaaaagtgcCCAAGTGACCATACGACAGG |
| JLG-251 | gggttaacgcgtaatccatgGACAGAGTGACAAGCGATCC |
| JLG-252 | gggttgccagagttaaaggaAAGTAAATTGCAGGGAACACC |
| JLG-253 | TCCTTTAACTCTGGCAACCC |
| JLG-254 | ATGTGATAACTCGGCGTATG |
| JLG-138 | CGAAGGCAGCAGTTTTTTGG |
| JLG-139 | ATAGAGATCCGATCAGACCAG |
| JLG-152 | TGCGAATTGTTTCATATTCAG |
| JLG-153 | GTTTTCTTCCTCTCTCATTGTTTC |
| JLG-297 | TACTGTTTTTTTCATCGGTCC |
| JLG-299 | gaaacaatgagagaggaagaaaac ATGTTTAAGATAAAGAAAAAGAAACTTTTTATAC |
| JLG-300 | ctggtctgatcggatctctat ACCTTGTTTTAGGCAAATGG |
| JLG-301 | ggaccgatgaaaaaaacagta ATGTTTAAGATAAAGAAAAAGAAACTTTTTATAC |
| JLG-302 | ctgaatatgaaacaattcgca ATGTTTAAGATAAAGAAAAAGAAACTTTTTATAC |
| JLG-303 | ccaaaaaactgctgccttcg ACCTTGTTTTAGGCAAATGG |
| JLG-304 | gaaacaatgagagaggaagaaaac GTGGATGCAATGACAAATAAAC |

*Appendix 3—table 3 continued on next page*

Appendix 3—table 3 continued

| Primer | Sequence[†] |
|--------|-------------|
| JLG-306 | ctggtctgatcggatctctat GGAACCATACGAATAACCCG |
| JLG-306 | ggaccgatgaaaaaaacagta GTGGATGCAATGACAAATAAAC |
| JLG-307 | ctgaatatgaaacaattcgca GTGGATGCAATGACAAATAAAC |
| JLG-308 | ccaaaaaactgctgccttcg GGAACCATACGAATAACCCG |
| JLG-453 | TGCGCTTGCGCTTGCGCTG |
| JLG-889 | gctagcagcgcaagcgcaagcgca ATGTTTAAGATAAAGAAAAAGAAACTTTTTATAC |
| JLG-890 | gctagcagcgcaagcgcaagcgca GTGGATGCAATGACAAATAAAC |
| JLG-891 | gaaacaatgagagaggaagaaaac GCTAAAGGCGAAGAACTGTTTAC |
| JLG-892 | ggaccgatgaaaaaaacagta GCTAAAGGCGAAGAACTGTTTAC |
| JLG-893 | ctgaatatgaaacaattcgca GCTAAAGGCGAAGAACTGTTTAC |
| JLG-894 | tgcgcttgcgcttgcgctgctagc TTTATACAGTTCATCCATGCC |
| JLG-977 | cagcgcaagcgcaagcgca ATGTCAGATCAATTTAACAGCC |
| JLG-978 | ctggtctgatcggatctctat TACCAAAAAAGCCATCACCC |
| JLG-979 | ccaaaaaactgctgccttcg TACCAAAAAAGCCATCACCC |
| JLG-980 | cagcgcaagcgcaagcgca GTGACCATGTTACGAAAAATAATC |
| JLG-981 | ctggtctgatcggatctctat TCTGAAGTCACTCCATATCCC |
| JLG-982 | ccaaaaaactgctgccttcg TCTGAAGTCACTCCATATCCC |
| JLG-1021 | cagcgcaagcgcaagcgca ATGATTCAAATGCCAAAAAAG |
| JLG-1022 | ctggtctgatcggatctctat TTTGGACAGGTAGAACGATG |
| JLG-1023 | ccaaaaaactgctgccttcg TTTGGACAGGTAGAACGATG |
| JLG-1024 | cagcgcaagcgcaagcgca ATGAAGCAGAATAAAAGAAAGCATC |
| JLG-1025 | ctggtctgatcggatctctat CATTCCTTTCTACTTCGTACGG |
| JLG-1026 | ccaaaaaactgctgccttcg CATTCCTTTCTACTTCGTACGG |
| JLG-1027 | cagcgcaagcgcaagcgca ATGAACCTTTTTTTCCTAGCTG |
| JLG-1028 | ctggtctgatcggatctctat CGCTAGAAAATGAGTATTCTCCTTC |
| JLG-1029 | ccaaaaaactgctgccttcg CGCTAGAAAATGAGTATTCTCCTTC |
| JLG-1030 | cagcgcaagcgcaagcgca ATGAAGATATCGAAACGAATGAAG |
| JLG-1031 | ctggtctgatcggatctctat TCTGCACTCCTTTATCCCTC |
| JLG-1032 | ccaaaaaactgctgccttcg TCTGCACTCCTTTATCCCTC |
| JLG-1033 | cagcgcaagcgcaagcgca ATGACAAGCCCAACCCGCAG |
| JLG-1034 | ctggtctgatcggatctctat CCATCTTAACGTTTGCAGGC |
| JLG-1035 | ccaaaaaactgctgccttcg CCATCTTAACGTTTGCAGGC |
| JLG-1036 | cagcgcaagcgcaagcgca ATGAGGAGAAATAAACCAAAAAAG |
| JLG-1037 | ctggtctgatcggatctctat AAGGTTTTGTAAATCAGTGCG |
| JLG-1038 | ccaaaaaactgctgccttcg AAGGTTTTGTAAATCAGTGCG |
| JLG-1039 | cagcgcaagcgcaagcgca TTGAACATCAAGAAATGTAAACAG |
| JLG-1040 | ctggtctgatcggatctctat TGGGTTTTTTCAGTATATTACGC |
| JLG-1041 | ccaaaaaactgctgccttcg TGGGTTTTTTCAGTATATTACGC |
| JLG-1042 | cagcgcaagcgcaagcgca ATGCGCATTTTCAAAAAAGCAG |
| JLG-1043 | ctggtctgatcggatctctat GATCACGGTTAAACTGACCC |
| JLG-1044 | ccaaaaaactgctgccttcg GATCACGGTTAAACTGACCC |
| JLG-1045 | cagcgcaagcgcaagcgca ATGAAAAAAAGCATAAAGCTTTATG |
| JLG-1046 | ctggtctgatcggatctctat CTAATTGTTGGAAGGTTCGAC |

*Appendix 3—table 3 continued on next page*

*Appendix 3—table 3 continued*

| Primer | Sequence[†] |
|--------|-------------|
| JLG-1047 | ccaaaaaactgctgccttcg CTAATTGTTGGAAGGTTCGAC |
| JLG-1048 | cagcgcaagcgcaagcgca ATGAAACGTCTTTTATCCACTTTG |
| JLG-1049 | ctggtctgatcggatctctat ATGAATTCCTTCACCGTGAC |
| JLG-1050 | ccaaaaaactgctgccttcg ATGAATTCCTTCACCGTGAC |
| JLG-1312 | gggttaacgcgtaatccatgACGGATAATCAGCATATCGG |
| JLG-1313 | gcctgagcgagggagcagaaGCAGAGGTGAGACAAGTGG |
| JLG-1314 | gcgttgaccagtgctccctgcTCTCCAGACCATCTCAAGTG |
| JLG-1315 | cacatttccccgaaaagtgcTCAATTCCAACAGAGATTGC |
| JLG-1330 | cagcgcaagcgcaagcgcaATGTTTGGAATTGGTGCTAG |
| JLG-1331 | ctggtctgatcggatctctatCACCTCTTCTATTGAACTCCC |
| JLG-1332 | ccaaaaaactgctgccttcgCACCTCTTCTATTGAACTCCC |
| JLG-1333 | cagcgcaagcgcaagcgcaATGTTTGCAAGGGATATTGG |
| JLG-1334 | ctggtctgatcggatctctatCCAGTTGTCATATAGGAACGTTC |
| JLG-1335 | ccaaaaaactgctgccttcgCCAGTTGTCATATAGGAACGTTC |
| JLG-1336 | cagcgcaagcgcaagcgcaATGTTTCAATCAACTGAAATCG |
| JLG-1337 | ctggtctgatcggatctctatCTCTTAGCATCTGTTTCCTCC |
| JLG-1338 | ccaaaaaactgctgccttcgCTCTTAGCATCTGTTTCCTCC |
| oER421 | ttctgctccctcgctcaggcggccgcATGAGAGAGGAAGAAAACGG |
| oER422 | cagggagcactggtcaacgctagcAATTGGGACAACTCCAGTG |

[†]In capital letters are shown the regions of the primer that anneal to the template. Homology regions for Gibson assembly are shown in italics.

