## [Decision Letter]

Thank you for submitting your article "Cell wall remodeling drives engulfment during *Bacillus subtilis* sporulation" for consideration by *eLife*. Your article has been reviewed by two peer reviewers, and the evaluation has been overseen by a guest Reviewing Editor and Naama Barkai as the Senior Editor. The reviewers have opted to remain anonymous.

The reviewers have discussed the reviews with one another and the Reviewing Editor has drafted this decision to help you prepare a revised submission.

Summary:

We find the manuscript interesting, insightful and well written. The experimental data provide a wealth of novel insights and the synthesis provided by the modeling scheme nicely fit engulfment dynamics of the wild-type and mutant phenotypes. Nevertheless, there are multiple points which need further clarifications both on the experimental and theoretical sides. We therefore recommend acceptance with major revisions that will answer the concerns of the reviewers and the Reviewing Editor.

Overview of the manuscript:

Previous works showed that peptidoglycan (PG) production occur at the leading edge of the engulfing membrane and that this leading edge of the mother-cell (MC) membrane seem to invade below the old cell wall made prior to asymmetric septation (Tocheva et al., 2013). In addition, reduced PG synthesis was also shown to affect engulfment (Meyer et al., 2010). The current work goes beyond the previous works and show that: 1) A block of PG synthesis halts engulfment completely. 2) PG synthesis at the leading edge (LE) is primary guided by PBPs localized to the leading edge at the forespore side. 3) SpoIIP localization to the LE is dependent on PG synthesis.

The authors present a biophysical model of PG synthesis which recapitulates the observed membrane dynamics of wild-type and mutant phenotypes by assuming that PG synthesis from the forespore (FS) is followed by degradation of the links between the old (pre-septation) and new (forespore dependent) PG layers, allowing the MC membrane to fill this gap.

Essential revisions:

The comments are both on the experimental and modeling parts of the manuscript.

*Experimental work:*

The major requests of the reviewers and the Reviewing Editor are the following:

1) PBP localization. Reviewer #1 expressed concerns regarding the mechanism of localization of PBPs. One of the options is that this localization is guided by other forespore specific proteins which interact with the leading edge. Specifically, as was raised during the discussion, we ask that the localization of PBPs will be studied in mutants of the FS-MC channel composed of spoIIQ, spoIIIAH and GerM.

2) Reviewer #2 suggests that real time monitoring of PBPs processive activity as was done for MreB (Domínguez-Escobar et al., 2011; Garner et al., 2011) would enable direct comparison with the model predictions.

*Modeling:*

The mathematical model has multiple underlying assumptions. We ask that you will further discuss these assumptions and extend the modeling scheme to further understand which of those is necessary. The major specific concerns raised are:

1) How does the model fit to the Gram positive envelope with its multiple layers, the presence of techioic acid and the unclear arrangement of PG strands (Reviewer #1)? Specifically, the model is very different than the one presented in Nguyen et al., 2015. Can the authors discuss these differences?

2) Discuss and simulate the validity of the assumption that the forespore-dependent synthesis of PG results in a difference between the new and old PG structure that enable specific breakage of the connecting peptide bond (Reviewer #1 and Reviewing Editor).

3) The presence of a tight insertion-degradation complex (IDC) is speculative. Compare simulation results of tight IDC activity (shown now) with simulations where synthesis and degradation are not coupled into a complex but are more weakly associated (as in Figure 2).

4) Further illuminate the functional importance of the 'make before break' model for this process (see further elaboration of this point below in the section "Comments on modeling raised during discussion").

Comments on modeling raised during discussion:

1) 'Make before break' and cell wall integrity. It is not clear to me whether the make before break process in the model is really needed for maintaining cell-wall integrity, as the model anyway assumes that the DMP complex is not effective in breaking the old cell wall and therefore does not jeopardize its integrity. It seems to me that the 'make' part is only necessary to ensure the specificity of the 'break' part. That is, to ensure that a forespore specific layer will be produced that allows the specific degradation of its connections with the old pre-forespore layer above it.

2) "Make just before break" vs. "make before break". Is localization to the LE critical or just more economic? It is not clear to me that the process would not have worked if the new layer would have been produced everywhere and then hydrolyzed specifically at the LE. It would be illuminating to see a simulation where it is assumed that there is no localization and the difference between cases discussed.

3) The role of the DMP complex in the simulations. The simulations seem only to show the making of the forespore inner layer of PG, but does not say anything about the interaction between this layer and the old layer and the corresponding mechanism of degradation of links between the two layers by the DMP complex. In effect, it seems that the DMP complex has no role in the IDC in the simulation. This might be OK, if one assumes that there is an IDC, but the authors claim in the Discussion that similar behavior would be observed if the two are not tightly linked. Can the authors present a more general model where spatial association is not tight (as shown in Figure 2 and discussed in the Discussion section)?

*Reviewer #1:*

The manuscript by Ojkic et al. presents a wealth of data on the mechanisms of endosporulation in *Bacillus subtilis*. In particular, they used fluorescence microscopy to observe the process of engulfment in the presence and absence of drugs inhibiting peptidoglycan synthesis. The data confirm previous studies concluding that both, peptidoglycan synthesis and hydrolysis are needed for membrane migration during engulfment. Multiple PBPs localized to the leading edge of engulfment. They present a model of how a biosynthetic complex and hydrolytic enzymes together facilitate engulfment by remodelling the peptidoglycan layer. Although a lot of data are presented in this impressive work I do have problems with the modelling. In my view the modelling goes too far and is quite speculative. Key aspects of the model are not supported by experimental data.

My specific points are as follows:

1) Introduction, statement about the "Gram-negative like PG layers in *Bacillus subtilis*" and modelling of the peptidoglycan. The architecture of PG is still a matter of debate, most data are available for *E. coli* and these support a single disordered layer made of relatively short glycan chains connected by peptide cross-links. However, although the Jensen lab hypothesized based on cryo-EM imaging that Gram-positive species stack multiple of such layers, there is not really good evidence that this model is correct. Other models have been proposed for example the one presented in Nguyen et al., 2015, which is quoted only for the glycan chain length but not for the *Bacillus* peptidoglycan model. The model presented in Nguyen et al., 2015 presents a more complicated arrangement of glycan chain bundles and was based on AFM images (Foster lab). The peptidoglycan from *B. subtilis* has significantly longer glycan chains than that from *E. coli*, and in *B. subtilis* the peptidoglycan is loaded with a significant amount of wall teichoic acid. Hence, we currently do not know the precise architecture of the peptidoglycan-wall teichoic acid in *B. subtilis*. The model presented here for the peptidoglycan architecture at the site of engulfment cannot be tested with any current technology and has therefore limited value.

2) Subsection “PG synthesis is essential for membrane migration”, first paragraph. Because fosfomycin and D-cycloserine failed to completely block polar division, they concluded that peptidoglycan might be obtained by recycling during starvation conditions. However, this is a quite speculative assumption which does not seem to be logical, because recycling requires peptidoglycan turnover, which occurs to significant extent only in growing bacteria. Does the mother cell grow during asymmetric septation, or where would the recycling material come from?

3) Discussion, first paragraph. It is not clear what is meant by the 'unique chemical composition of the peptide bridges' that are recognized by DMP. This would imply that the same PBPs, which were found at the leading edge of engulfment and which synthesize the peptidoglycan of the lateral wall or septum during vegetative growth, produce peptide bridges with different composition when they are active during engulfment. This is a highly speculative assumption.

4) Discussion, second paragraph. The PG-insertion-degradation complex (IDC). This is another speculation that is not based on evidence, as they do not present any interaction data between the different peptidoglycan enzymes (PBPs and hydrolases) and other engulfment proteins.

*Reviewer #2:*

The manuscript describes several experimental findings that advance understanding of engulfment during Bacillus sporulation. The new insights are used to formulate a mathematical model that reproduces experimentally observed engulfment phenotypes. Together, the experimental and modeling results are an important contribution since engulfment is crucial for endospore formation but a mechanistic understanding has been lacking.

The main experimental findings are 1) peptidoglycan synthesis appears to be essential for migration of the leading edge (LE) of the engulfing mother cell membrane and for localization of SpoIIP (a protein in a peptidoglycan degradation complex) to the LE, based on results obtained with inhibitors of peptidoglycan synthesis, and 2) peptidoglycan-binding proteins (PBPs), which synthesize peptidoglycan, localize to the LE, in most cases only if the PBP is expressed in the forespore. Based on these findings, the authors propose that peptidoglycan synthesis and degradation by forespore PBPs and the mother cell SpoIIP-containing complex, respectively, causes the junction between septal peptidoglycan and the lateral cell wall to move, creating space into which the LE of the mother cell membrane moves by entropic forces.

The authors formulate a model based on the "template mechanism" of vegetative cell growth, in which existing glycan strands serve as a "template" for synthesis and peptide cross-linking of a new glycan strand prior to degradation of "old" peptide cross-links and perhaps some of the "old" glycan strands. Dynamic simulations with the model produce engulfment with timing, and with forespore area and volume, that match the experimental observations. Simulations in which the probability of the modeled "insertion-degradation complex" initiating and continuing polymerization at glycan ends is too low result in asymmetric engulfment, as observed experimentally when inhibitors of peptidoglycan synthesis are added.

I support publication in *eLife* based on the fundamental biological insight provided, the convincing data, and the excellent presentation which is suitable for a broad audience.

That said, the manuscript could be strengthened as follows:

1) Provide direct evidence for peptidoglycan synthesis at the LE (e.g., using fluorescent D-amino acids). The data do not completely rule out a mechanism involving only degradative remodeling of the lateral cell wall to create the germ cell wall, if the peptidoglycan synthesis inhibitors used unexpectedly inhibit degradation.

2) Track forespore-expressed GFP-PBP fusions (as for GFP fusions to MreB isoforms in Domínguez-Escobar et al., 2011and Garner et al., 2011). If the predicted circumferential motions were observed, and, if it were possible to measure their number and speed, predictions of the modeling made in the last paragraph of Results could be tested.

3) For completeness, do parallel experiments on localization of SpoIID and SpoIIM, to those reported on SpoIIP, since the three proteins are expected to form a complex.

4) Clarify whether the probability of initiating glycan polymerization from an end defect (*p*_def_ in subsection “A biophysical model to describe leading edge migration”) is different from the probability of inserting new glycan from an old glycan end and repairing the end defect (prep in Figure 3 legend).

---

## [Author Response]

Essential revisions:

*The comments are both on the experimental and modeling parts of the manuscript.*

Experimental work:

*The major requests of the reviewers and the Reviewing Editor are the following:*

*1) PBP localization. Reviewer #1 expressed concerns regarding the mechanism of localization of PBPs. One of the options is that this localization is guided by other forespore specific proteins which interact with the leading edge. Specifically, as was raised during the discussion, we ask that the localization of PBPs will be studied in mutants of the FS-MC channel composed of spoIIQ, spoIIIAH and GerM.*

We have determined the localization of GFP-PonA and GFP-PbpA produced in the forespore from the *spoIIR* promoter, in mutants lacking SpoIIQ, SpoIIIAH or GerM. Bright foci coincident with the leading edge of the engulfing membrane are still observed in the three mutant backgrounds, suggesting that the Q-AH transenvelope complex is not required for forespore PBP localization.

To further investigate the localization mechanism, we have examined forespore PBP localization in additional mutant backgrounds. We have used strains lacking SpoIIB or SpoIVFAB, which are required for septal localization of the DMP complex (Aung et al., 2007). GFP-PonA and GFP-PbpA still track the leading edge of the engulfing membrane in both backgrounds. We have also tested PBP localization in mutants lacking SpoIID, SpoIIM and SpoIIP. In these backgrounds, engulfment membrane migration is blocked and the septal membrane bulges towards the mother cell cytoplasm through the middle of the septum. Interestingly, GFP-PonA and GFP-PbpA form bright foci at the intersection between the septum and the lateral cell wall. A similar localization pattern is observed σ^E-^ mutants, which lack mother cell-specific gene expression. These results leave two possibilities to explain for the localization of forespore PBPs. First, it is possible that PBPs directly recognize the junction between the septal peptidoglycan and the lateral cell wall, and track it as it moves around the forespore. Second, forespore PBP localization might rely on hitherto unknown forespore-specific factors.

These results are now presented in Figure 2—figure supplement 2 and mentioned in the last paragraph of the subsection “PG biosynthetic machinery tracks the leading edge of the engulfing membrane from the forespore”.

*2) Reviewer #2 suggests that real time monitoring of PBPs processive activity as was done for MreB (Domínguez-Escobar et al., 2011; Garner et al., 2011) would enable direct comparison with the model predictions.*

We thank the reviewer for suggesting this experiment. To address this point we have constructed new GFP fusions to the three MreB isoforms in *Bacillus subtilis* (MreB, Mbl and MreBH), since our GFP-PBP fusions photobleached quickly and were not bright enough to perform tracking experiments. MreB, Mbl and MreBH displayed the same localization pattern that most PBPs, localizing to the LE of the engulfing membrane when produced in the forespore, but not when produced in the mother cell. This data has been incorporated to Figure 2 and Figure 2—figure supplement 1. We have monitored the movement of GFP-MreB specifically produced in the forespore from *spoIIQ* promoter by TIRF microscopy. Forespore GFP-MreB foci rotate around the forespore, coincident with the leading edge of the engulfing membrane, with speeds equivalent to those determined in vegetative cells (Garner et al., 2011). This data has been incorporated to Figure 2 (Figure 2), and included in a new video (Video 2). The results are explained in the first paragraph of the subsection “PG biosynthetic machinery tracks the leading edge of the engulfing membrane from the forespore”, and we made reference to them in the last paragraph of the Results section, and in the Discussion section.

Modeling:

*The mathematical model has multiple underlying assumptions. We ask that you will further discuss these assumptions and extend the modeling scheme to further understand which of those is necessary. The major specific concerns raised are:*

*1) How does the model fit to the Gram positive envelope with its multiple layers, the presence of techioic acid and the unclear arrangement of PG strands (reviewer #2)? Specifically, the model is very different than the one presented in Nguyen et al., 2015. Can the authors discuss these differences?*

Our model deals with the formation of the cell wall that surrounds the spore and the cell wall remodeling at the leading edge, and does not require modeling the precise structure of the outer cell wall. This is now mentioned in the first paragraph of the subsection “A biophysical model to describe leading edge migration”, and allows us to simplify simulations, avoid uncertainties about the structure of the outer cell wall, and focus on the essential aspects of engulfment. Our template mechanism is easiest to envision if the glycan strands spiral in loops around the long axis of the cell similar to Gram-negative cell wall since enzymes have tracks to move on (similar to how MreB move; in fact, through the working of the cell-wall remodeling enzymes we essentially predict this cell-wall structure.) However, the model that the lateral cell wall is organized in bundles of PG strands (Hayhurst et al., (2008), Ref. 39) is also compatible with our model, since the cables could still provide tracks for the movement of the PG biosynthetic machinery. Thus, due to the ongoing uncertainty in the cell-wall structure of *B. subtilis*, we decided to remove the phrase “multiple layers of Gram-negative-like PG“. We also now point out that unlike the Gram-negative cell wall, the Gram-positive cell wall contains significant amounts of teichoic acids in the Introduction (third paragraph). However, because wall-teichoic acids are largely absent in spores [Chin et al.,1968; Johnstone et al., 1982] (Chin, Younger and Glaser, 1968; Johnstone, Simion and Ellar, 1982) and teichoic acids are not required for engulfment (JLG and KP unpublished data), we do not consider them in our model.

The model in Nguyen et al., 2015 represents a detailed molecular-level simulation (although coarse-grained) of cell-wall growth in vegetative Gram-negative bacteria (*E. coli*). The overlap is that as in model of Laporte et al., 2012 the local enzymatic coordination is enough to explain micro scale PG transformations. However, we only used the PG elastic parameter from their molecular dynamic simulations (*k*gly and *k*pep, see Figure 4). We added clarifying comments in the section on “Langevin simulations reproduce observed phenotypes”.

*2) Discuss and simulate the validity of the assumption that the forespore-dependent synthesis of PG results in a difference between the new and old PG structure that enable specific breakage of the connecting peptide bond (Reviewer #1 and Reviewing Editor).*

Our model already postulates that DMP specifically targets peptide cross links joining new and old PG. We now discuss the two main possible explanations for how the peptide crosslinks joining the newly synthesized septal peptidoglycan and the lateral cell walls could be specifically recognized based on architectural and chemical features (see Discussion section):

DMP might recognize the junction between the septal PG and the lateral PG based on some specific feature of the PG architecture at this site, such as having a peptide bond/glycan strand projecting into a different plane than the rest of the PG. We simulated the possibility that PG degradation also cuts penultimate peptide connections (peptide connections in different planes) with probability *p*pcut (Figure 4—figure supplement 5). For relatively small *p*pcut = 0.1 an irregular peptidoglycan meshwork is formed. Even more, since antibiotic treatment dislocates DMP degradation machinery (Figure 2, Figure 2—figure supplement 3) we explored the possibility that dislocated DMP randomly cuts old germ cell wall peptides with constant degradation rate *p*rpep. In this scenario, the irregular peptidoglycan network protrudes towards the mother cell with apparent volume increase while the leading edge remains still (Figure 4—figure supplement 5).

*3) The presence of a tight insertion-degradation complex (IDC) is speculative. Compare simulation results of tight IDC activity (shown now) with simulations where synthesis and degradation are not coupled into a complex but are more weakly associated (as in Figure 2).*

Thank you for this suggestion. We implemented simulations without a tight IDCs, in which DMP degrades junctional peptide bonds between old outer cell wall and newly synthesized germ cell wall with a slight time delay after new PG is made (explained in supplementary information with results shown in Figure 4—figure supplement 5, Video 6). As long as DMP does not make significant amounts of errors in terms of degrading accidentally either the outer cell wall or the septal peptidoglycan (Figure 4—figure supplement 5), this mechanism supports engulfment as well as that synthesis and degradation are tightly coupled. We comment on these important new results in the Discussion section.

*4) Further illuminate the functional importance of the 'make before break' model for this process (see further elaboration of this point below in the section "Comments on modeling raised during discussion").*

We clarified this point in response to the next question and in the Discussion (fourth paragraph).

Comments on modeling raised during discussion:

*1) 'Make before break' and cell wall integrity. It is not clear to me whether the make before break process in the model is really needed for maintaining cell-wall integrity, as the model anyway assumes that the DMP complex is not effective in breaking the old cell wall and therefore does not jeopardize its integrity. It seems to me that the 'make' part is only necessary to ensure the specificity of the 'break' part. That is, to ensure that a forespore specific layer will be produced that allows the specific degradation of its connections with the old pre-forespore layer above it.*

This is a good point. As the reviewers point out, during engulfment, the primary contributions of the ‘make before break’ principle might well be to confer directionality to membrane migration by producing a forespore specific substrate for the mother cell enzymes that degrade PG (DMP) and by ensuring the robust attachment of the septal cell wall to the lateral cell wall. We have clarified this point in the Discussion section.

*2) "Make just before break" vs. "make before break". Is localization to the LE critical or just more economic? It is not clear to me that the process would not have worked if the new layer would have been produced everywhere and then hydrolyzed specifically at the LE. It would be illuminating to see a simulation where it is assumed that there is no localization and the difference between cases discussed.*

As described in response to item 3 of essential revisions, we conducted simulations where DMP degrades specific junctional peptide bonds with a delay. Indeed, as long as these bonds are clearly labeled as DMP substrate, such degradation could occur after the new cell wall fully encloses the forespore. However, bacterial cells recycle peptidoglycan fragments produced during growth and this might be particularly important during starvation – making a large amount of PG without recycling might constitute a major energetic burden. Hence, one could consider naming this a “make just before break” model, and we discuss this now in the Discussion section (fourth paragraph).

*3) The role of the DMP complex in the simulations. The simulations seem only to show the making of the forespore inner layer of PG, but does not say anything about the interaction between this layer and the old layer and the corresponding mechanism of degradation of links between the two layers by the DMP complex. In effect, it seems that the DMP complex has no role in the IDC in the simulation. This might be OK, if one assumes that there is an IDC, but the authors claim in the Discussion that similar behavior would be observed if the two are not tightly linked. Can the authors present a more general model where spatial association is not tight (as shown in Figure 2 and discussed in the Discussion section)?*

In our model, we assume only that the peptide cross-link between the old and new cell wall is removed according to our schematic Figure 3 by DMP. We discuss this point in the text, acknowledging more complicated versions (see Figure 3—figure supplement 1) in which the entire junctional strand is removed (peptide and glycan), and we have extended Figure 3 to show these cross-links. As discussed above, we also include a model that lacks tightly coupled IDCs to support this point.

Reviewer #1:

*[…] Although a lot of data are presented in this impressive work I do have problems with the modelling. In my view the modelling goes too far and is quite speculative. Key aspects of the model are not supported by experimental data.*

*My specific points are as follows:*

*1) Introduction, statement about the "Gram-negative like PG layers in Bacillus subtilis" and modelling of the peptidoglycan. The architecture of PG is still a matter of debate, most data are available for E. coli and these support a single disordered layer made of relatively short glycan chains connected by peptide cross-links. However, although the Jensen lab hypothesized based on cryo-EM imaging that Gram-positive species stack multiple of such layers, there is not really good evidence that this model is correct. Other models have been proposed for example the one presented in Nguyen et al., 2015, which is quoted only for the glycan chain length but not for the Bacillus peptidoglycan model. The model presented in Nguyen et al., 2015 presents a more complicated arrangement of glycan chain bundles and was based on AFM images (Foster lab). The peptidoglycan from B. subtilis has significantly longer glycan chains than that from E. coli, and in B. subtilis the peptidoglycan is loaded with a significant amount of wall teichoic acid. Hence, we currently do not know the precise architecture of the peptidoglycan-wall teichoic acid in B. subtilis. The model presented here for the peptidoglycan architecture at the site of engulfment cannot be tested with any current technology and has therefore limited value.*

We acknowledge the ongoing uncertainty about the organization of the *B. subtilis* lateral cell wall, and we have removed the “Gram-negative like PG layers” statement in the Introduction (see response to the first question of “Essential reviews of modelling” for more details). It is important to note that our model deals with the movement of the junction between the septal peptidoglycan and the lateral cell wall around the forespore and it does not depend on knowing the precise architecture of the lateral cell wall. We modeled the extended septal peptidoglycan as a single peptidoglycan layer since it is only 2 nm thick (Tocheva et al., 2013), and hence cannot accommodate 50 nm wide cables.

The model only depends on *B. subtilis* cells being able to produce septa at various places along the lateral cell wall in addition to midcell, as is supported by the variable locations of septa in the minCD and DivIVA mutants (Gregory and Pogliano, 2009; Edwads and Errington, 1997). Thus, as long as docking sites are available to make new connections between the septal PG and the lateral cell wall, the specific architecture of the lateral cell wall is irrelevant for our model. We did not include teichoic acids in the model since they are absent in spores (Chin et al., 1968; Johnstone et al.1982).

In our opinion, the model has intrinsic value because it easily conceptualizes how cells could move the engulfing membrane around the forespore, by simply moving the junction between the septum and the lateral wall using known enzymatic activities. As a result, the model has predictive value, even if it ultimately must be revised to accommodate new data concerning the architecture of PG. Emerging experimental techniques, such as FIB-CryoEM, subtomogram averaging, and single molecule tracking have the potential to provide new insight into the specific organization of the peptidoglycan at the leading edge of the engulfing membrane and constitute a powerful direction of future research. We added this to the future outlook at the end of the Discussion section.

*2) Subsection “PG synthesis is essential for membrane migration”, first paragraph. Because fosfomycin and D-cycloserine failed to completely block polar division, they concluded that peptidoglycan might be obtained by recycling during starvation conditions. However, this is a quite speculative assumption which does not seem to be logical, because recycling requires peptidoglycan turnover, which occurs to significant extent only in growing bacteria. Does the mother cell grow during asymmetric septation, or where would the recycling material come from?*

We thank this reviewer for raising this interesting point. It is important to note that during sporulation, some cells are still growing and dividing, which may release PG precursors into the culture that could be used by sporulating cells. In addition, the continued polar septation of antibiotic-treated cells might be supported by PG that is released by lysis of non-sporulating cells in the culture either via the SDP toxin [Gonzalez-Pastor JE et al., Science 301, 510 (2003); Straight PD, Kolter R, Annu Rev Microbiol 63, 99 (2009), Lamsa A et al. Mol Micobiol 84, 486 (2012)] or due to antibiotic treatment, since fosfomycin and D-cycloserine treated cells also lyse (Lamsa et al., 2016). Engulfment might require much less PG than cell division, in which case SpoIIDMP activity might provide enough material.

*3) Discussion, first paragraph. It is not clear what is meant by the 'unique chemical composition of the peptide bridges' that are recognized by DMP. This would imply that the same PBPs, which were found at the leading edge of engulfment and which synthesize the peptidoglycan of the lateral wall or septum during vegetative growth, produce peptide bridges with different composition when they are active during engulfment. This is a highly speculative assumption.*

We provide details about this in the second point of the “Essential revisions of modeling”, and have clarified the text.

*4) Discussion, second paragraph. The PG-insertion-degradation complex (IDC). This is another speculation that is not based on evidence, as they do not present any interaction data between the different peptidoglycan enzymes (PBPs and hydrolases) and other engulfment proteins.*

We have conducted new simulations in which PG synthesis and degradation are uncoupled. Additional details can be found in the third item of “Essential revision of modeling”, and we have clarified the text.

Reviewer #2:

*[…] I support publication in eLife based on the fundamental biological insight provided, the convincing data, and the excellent presentation which is suitable for a broad audience.*

*That said, the manuscript could be strengthened as follows:*

*1) Provide direct evidence for peptidoglycan synthesis at the LE (e.g., using fluorescent D-amino acids). The data do not completely rule out a mechanism involving only degradative remodeling of the lateral cell wall to create the germ cell wall, if the peptidoglycan synthesis inhibitors used unexpectedly inhibit degradation.*

Thank you for clarifying this issue in the discussion between editors and reviewers. Indeed, the Pogliano lab studied PG synthesis by fluorescent D-amino acids and fluorescently labeledantibiotics that reveal PG precursor localization, and we here extend this study to the localization of individual PBPs and to the penicillin V derivative bocillin-FL as explained in the subsection “PBPs accumulate at the leading edge of the engulfing membrane” (Tocheva et al., 2013; Reith and Meyer, 2011).

*2) Track forespore-expressed GFP-PBP fusions (as for GFP fusions to MreB isoforms in Domínguez-Escobar et al., 2011and Garner et al., 2011). If the predicted circumferential motions were observed, and, if it were possible to measure their number and speed, predictions of the modeling made in the last paragraph of Results could be tested.*

Thank you for suggesting this experiment. We have tracked GFP-MreB specifically produced in the forespore after polar septation. Please, see the response to the second “Essential revision of experimental work” for details.

*3) For completeness, do parallel experiments on localization of SpoIID and SpoIIM, to those reported on SpoIIP, since the three proteins are expected to form a complex.*

*4) Clarify whether the probability of initiating glycan polymerization from an end defect (p_def_ in subsection “A biophysical model to describe leading edge migration”) is different from the probability of inserting new glycan from an old glycan end and repairing the end defect (prep in Figure 3 legend).*

We have determined the localization of GFP-SpoIID and GFP-SpoIIM in sporangia treated with cephalexin and bacitracin. In both cases the fusion proteins delocalized upon antibiotic treatment, similar to GFP-SpoIIP, although neither protein localizes as well as SpoIIP. Those results have been included in Figure 2—figure supplement 3. We now consistently use *p*pro and *p*rep (as defined in Figure 3). We thank reviewer for pointing to this.